



# Salinity and Stratification at the Sea Ice Edge (SASSIE): An oceanographic field campaign in the Beaufort Sea

Kyla Drushka[1], Elizabeth Westbrook[2], Frederick M. Bingham[2], Peter Gaube[1], Suzanne Dickinson[1*], Severine Fournier[3*], Viviane Menezes[4*], Sidharth Misra[3*], Jaynise Perez[1*], Edwin J. Rainville[1*], Julian J. Schanze[5*], Carlyn Schmidgall[1,6*], Andrey Shcherbina[1*], Michael Steele[1*], Jim Thomson[1*], Seth Zippel[7*]

[1] Applied Physics Laboratory, University of Washington, Seattle WA, 98105, United States
[2] Department of Physics & Physical Oceanography, University of North Carolina Wilmington, Wilmington NC, 28412, United States
[3] Jet Propulsion Laboratory, California Institute of Technology, Pasadena, CA 91109, USA
[4] Woods Hole Oceanographic Institute, Falmouth, MA 02543, United States
[5] Earth and Space Research, Seattle, WA, 98105, United States
[6] School of Oceanography, University of Washington, Seattle WA, 98105, United States
[7] Oregon State University College of Earth Ocean and Atmospheric Sciences, Corvallis, OR, 97331, United States
*These authors listed in alphabetical order

*Correspondence to*: Peter Gaube (pgaube@uw.edu)

**Abstract**. As our planet warms, Arctic sea ice coverage continues to decline resulting in complex feedbacks with the climate system. The core objective of NASA's Salinity and Stratification at the Sea Ice Edge (SASSIE) mission is to understand how ocean salinity and near-surface stratification affect upper ocean heat content, and thus sea ice freeze and melt. SASSIE specifically focuses on the formation of Arctic sea ice in autumn. The SASSIE field campaign in 2022 collected detailed observations of upper ocean properties and meteorology near the sea ice edge in the Beaufort Sea using ship-based and piloted and drifting assets. The observations collected during SASSIE include vertical profiles of stratification up to the sea surface, air-sea fluxes, and ancillary measurements that are being used to better understand the role of salinity in coupled Arctic air-sea-ice processes. This publication provides a detailed overview of the activities during the 2022 SASSIE campaign and presents the publicly available datasets generated by this mission, inducing an accompanying repository that highlights the numerical routines used to generate the figures shown in this manuscript.

# 1 Introduction



## 1.1 Background

Sea ice extent in the Arctic Ocean has declined dramatically over the past decades. As a result of climate change, autumn ice advance is slower and occurs later, while summer ice retreat is faster and occurs earlier (Stroeve et al., 2014; Stroeve and Notz, 2018). The result is a lengthening open-water period each year, leading to changes in air-sea heat and momentum fluxes, the freshwater cycle, surface albedo feedbacks, primary production, and regional and global climate as well as human and ecological health (Lannuzel et al., 2020). Understanding the dynamics that govern the spatial and temporal patterns of sea ice formation is critical to understanding and predicting the impacts of the changing arctic cryosphere.

Salinity controls stratification in the cold Arctic Seas, enabling the uniquely polar phenomenon of colder, less-dense fresher waters situated above warmer, denser saltier waters. Variations in upper ocean salinity and the resulting stratification modulate the surface ventilation of stored subsurface heat, which can affect upper ocean temperature and thus sea ice formation and melt (e.g., Smith et al. 2018). During summer, melting sea ice leaves a layer of fresh, cold water at the ocean surface (Dewey et al, 2017). By mid-September, sea ice extent reaches a minimum: melting stops, and the ice edge begins to advance. The freshwater deposited by melting sea ice during its seasonal retreat may increase near-surface stratification, suppressing upward mixing of heat from the warmer subsurface waters and thereby enhancing surface cooling, as shown by Crews et al. (2022). Increased stratification from melted sea ice may precondition the ocean for autumn sea ice formation, but there are very few in situ ocean and sea ice observations of this connection.

Salinity and Stratification at the Sea Ice Edge (SASSIE) is a NASA physical oceanography mission that aims to clarify the role of salinity and upper ocean stratification in the Arctic Ocean. The primary goal of SASSIE is to better understand how the salinity anomalies generated by summer ice melt in the Western Arctic evolve in space and time and ultimately how they affect the upper ocean structure and the formation of sea ice in the early fall. This paper describes data collected during the SASSIE field campaign that took place in August-October 2022 in the Beaufort Sea, including in situ measurements collected from a ship and a suite of uncrewed and drifting platforms as well as remote sensing measurements collected from an aircraft. Satellite measurements and model simulations are also a crucial part of the SASSIE mission data but are not addressed here.

## 1.2 Campaign overview

The SASSIE campaign focused on capturing three regimes in the Beaufort Sea, a region characterized by strong near-surface salinity stratification. Late summer sea ice melt was sampled with a fleet of four Wave Gliders that traveled progressively northward from mid-August to early September as the sea ice retreated (Figure 1a, b, and Table 1). SASSIE's intensive observing period, from early September to early October, centered around the ship- and aircraft-based campaign during the transition between the summer melt season and fall freeze-up (Figure 1c, d). This main phase of the campaign focused on the

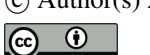



65    region of the Beaufort Sea bounded by the shelf-break to the south and the sea ice edge to the north, between 154°W and
145°W (Figure 2).   Following the intensive ship and airborne campaigns, the early part of the freeze-up was sampled with a
number of floats and drifters that remained in the water after the ship departed.

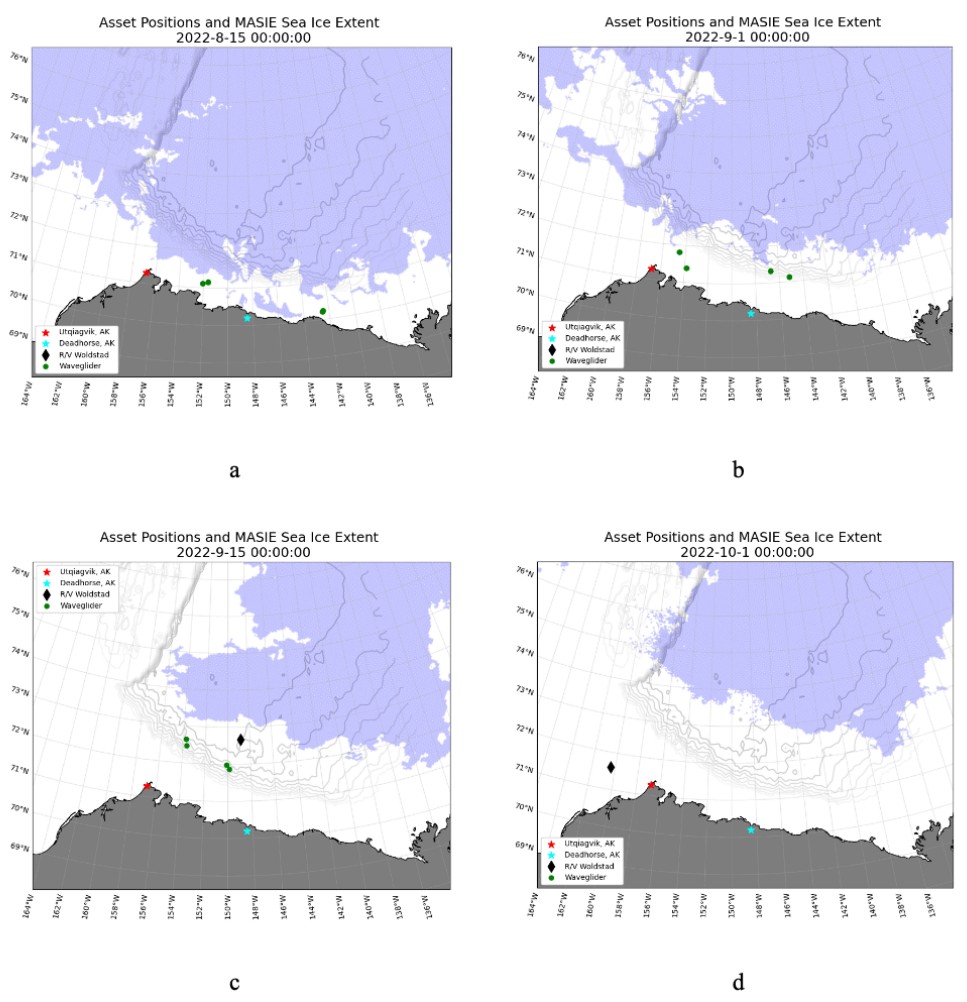

70

**Figure 1: Sea ice extent from the MASIE-NH product (US National Ice Center, 2010) for (a) 15 August, (b) 1 September, (c) 15 September, and (d) 1 October 2022. The positions of the RV Woldstad (black diamond) and the Wave Gliders (green circle) on each date are shown. Bathymetry contours from 1000 to 6000 m (gray lines) from the ETOPO2 product (Smith & Sandwell v. 8.2: 1/30-degree topography and bathymetry) are shown for reference.**

Earth System
Science
Data

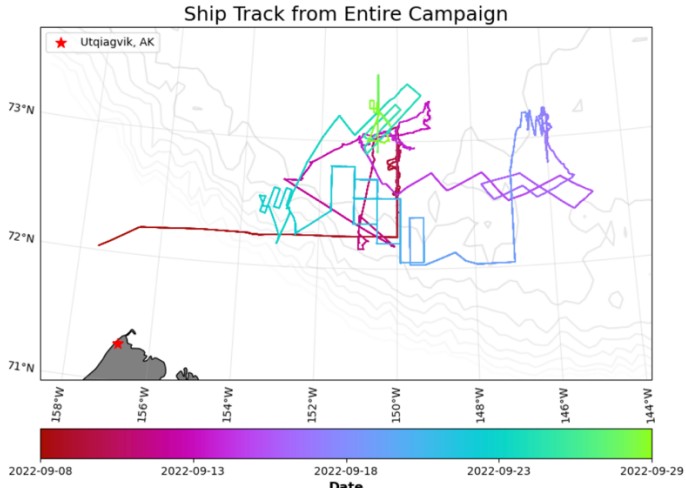

**Figure 2 The track of the RV *Woldstad* throughout the SASSIE campaign, colored by date. Bathymetry data (grey lines) from 1000 to 6000 m at an interval of 300m added for reference.**

The major daily activities of the SASSIE campaign are summarized in Table 1. An animated depiction of the platforms is included in the supplemental materials.

| Date (2022) | Event |
|---|---|
| **August** | |
| 12-14 | Wave Glider deployments from RV *Ukpik*. |
| 28-29 | RV *Woldstad* mobilization in Homer, AK. |
| 30 - Sept 5 | RV *Woldstad* transit from Homer, AK to Nome, AK with only crew aboard. |
| **September** | |
| 5 | RV *Woldstad* departs Nome with 12 scientists and 6 crew. Shipboard sensors begin collecting data. |
| 7 | RV *Woldstad* rounds Point Barrow. |
| 8 | RV *Woldstad* enters the study area and begins play #1: ice and open water survey. Four ALTO floats and 1 HydroBuoy deployed. |
| 9 | Five SWIFTs, 2 HydroBuoys deployed. |
| 10 | Under Ice Float deployed. JetSSP deployment #1. |
| 11 | Four Wave Gliders recovered. One ALAMO float and 1 HydroBuoy deployed. JetSSP deployment #2. Aircraft flight #1. |



| 12 | Four Wave Gliders redeployed and 5 SWIFTs recovered. |
|---|---|
| 13 | Two ice stations. One HydroBuoy deployed. Play #1 ends as the ship exits heavy ice cover. |
| 14 | Play #2 (zig-zag open water survey) begins. Aircraft flight #2. |
| 15 | One ALAMO float and 2 HydroBuoys deployed. |
| 16 | Play #2 ends. Play #3 (ice survey) begins. Aircraft flight #3. |
| 17 | 1 HydroBuoy deployed. Two SWIFTs deployed and recovered. JetSSP deployment #3. Aircraft flight #4 |
| 18 | Two SWIFTs deployed and recovered. Aircraft flight #5. Play #3 ends. |
| 19 | Wave Glider 245 recovered. Play #4 (drifter-following survey boxes) begins. One ALAMO, 2 HydroBuoys, and 2 SWIFTs deployed. |
| 20 | JetSSP deployment #4. |
| 21 | Wave Glider 245 redeployed. JetSSP deployment #5. One SWIFT deployed. |
| 22 | Wave Gliders 153 and 130 recovered. One SWIFT deployed. |
| 23 | Play #4 ends. Four SWIFTs recovered. |
| 24 | Wave Gliders 153 and 130 redeployed. |
| 25 | Wave Glider 247 recovered and redeployed. Play #5 (Repeated ice-open water transect) begins. Two HydroBuoys, 2 SWIFTs, and 1 ALAMO float deployed. JetSSP deployment #6. |
| 26 | JetSSP survey #7. One SWIFT deployed. |
| 27 | One SWIFT deployed and recovered. |
| 28 | Four Wave Gliders and 3 SWIFTs recovered. |
| 29 | Play #5 ends. Transit to Nome, AK begins. |
| **October** | |
| 1 | Air-sea flux sensors turned off. |
| 2 | Salinity snake recovered. |
| 3 | RV *Woldstad* arrives in Nome, AK. Science party disembarks. Ship transits to Homer with only crew aboard. |
| 10 | RV *Woldstad* demobilization in Homer, AK. |

80    **Table 1. A timeline of SASSIE events. Dates are local time. The plays referred to in the table are described in Sect. 3.**



A major objective of the SASSIE mission is to deliver quality-controlled data products to NASA's Physical Oceanography Distributed Active Archive Center (PO.DAAC) in order to foster broad utilization of the data within the broader scientific community. This paper outlines the campaign and describes the 15 SASSIE datasets archived at the PO. DAAC and available at https://podaac.jpl.nasa.gov/SASSIE. Python notebooks to download, read and visualize the data are available at the SASSIE's Data Vis GitHub page: https://doi.org/10.5281/zenodo.8308513. Sect. 2 describes the data and processing for each of the platforms deployed during SASSIE; Sect. 3 details the five "plays" that distinguish different periods of the research cruise; Sect. 4 gives information about how SASSIE data can be accessed; and Sect. 5 gives definitions of the abbreviations used in this paper.

## 2 Data

### 2.1 Research vessel and ship-deployed Instruments

The ship-based sampling of the SASSIE program was carried out on the RV *Woldstad* (Figure 3), a 121' deep-draft steel vessel operated by Support Vessels of Alaska. This section describes all measurements collected from the *Woldstad*. A thermosalinograph (TSG) and S-band navigation radar are permanently installed on the vessel, while the SASSIE team supplied all other shipboard instruments.

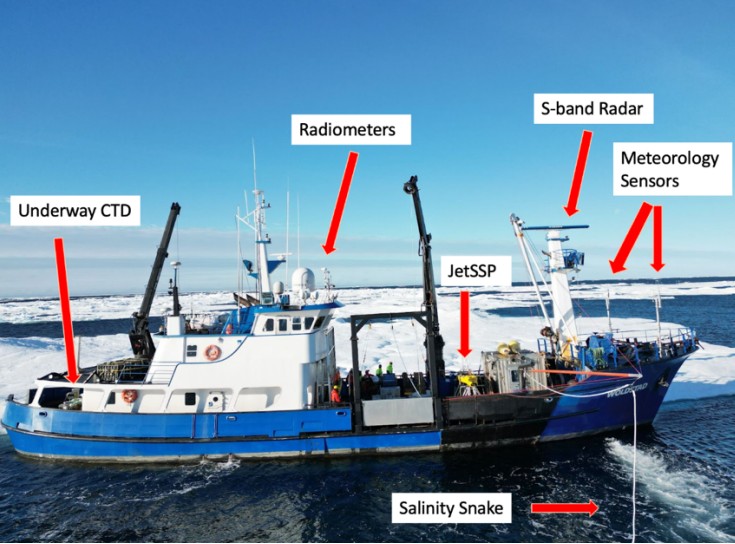

**Figure 3. The RV *Woldstad*. The photo was taken during the ice station on 13 September 2022. Note: 1) the boom for the salinity snake protruding from the starboard side of the ship in the foreground; 2) two air-sea flux masts on the bow; 3) the radiometers above the bridge; 4) the JetSSP ready for deployment on the foredeck; 5) uCTD winch on the stern. 7) S-band radar antenna on the mast.**



### 2.1.1 Thermosalinograph (Drushka, 2023a)

The shipboard TSG system pumped water from an intake on the ship's hull at around 4 m depth through a SBE38 temperature sensor, a vortex debubbler, and finally a SBE21 SeaCAT TSG. Temperature and conductivity data were logged every minute using SeaSave software. Temperature measurements were made from both the SBE21 and SBE38 instruments: the SBE21
temperature measurement is only used for the salinity computation; the SBE38 temperature more accurately reflects the ocean temperature because the sensor measures seawater prior to it going through the TSG.

Temperature data were quality controlled by removing values greater than 15°C (<0.001% of data) and spikes, defined as anomalies larger than 0.3°C over 1 minute (<0.005% of data). Salinity data were quality controlled by removing values smaller
than 22 (<0.03% of data) and negative spikes, defined as anomalies larger than –0.08 over 1 minute (<0.5% of data). Comparisons of salinity measurements from the TSG and from the three CTD sensors on the JetSSP (see section 2.2.2), during periods while the upper ocean was well-mixed and all JetSSP sensors agreed, indicate that the ship's TSG had a –0.05 (fresh) bias. This bias was corrected in the final data product.

The large fluctuations observed in the temperature and salinity time series (Figure 4) primarily reflect spatial variability captured as the ship moved between open regions south of the ice edge and areas with low to moderate ice cover. Surface waters in the SASSIE domain were typically warmer and saltier in open water (1 to 2.5°C, 26 to 28) and cooler and fresher within the ice (-1 to 0°C, <26) within the ice (Figure 5). An exception is the strong variability seen from 19-23 September, when the ship was in open water and sampled a region with strong fronts (Figure 4).



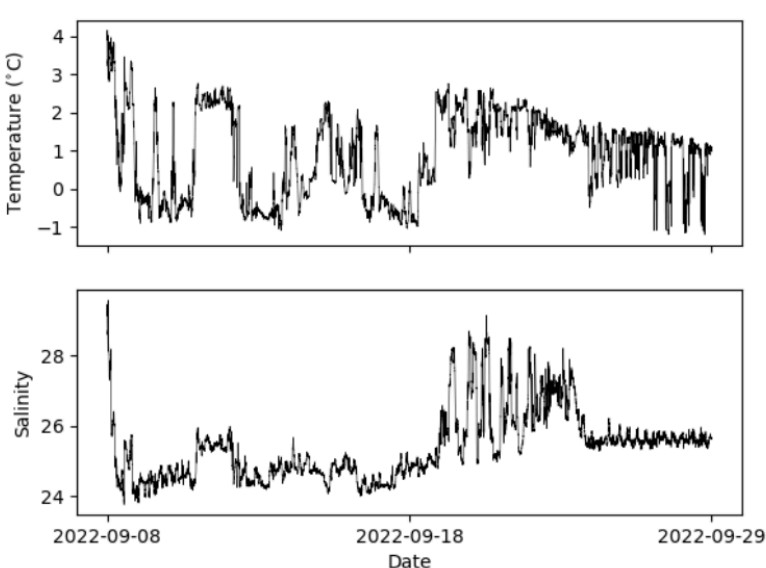

**Figure 4. Temperature (top) and salinity (bottom) records from the ship's thermosalinograph (TSG) during the SASSIE campaign.**

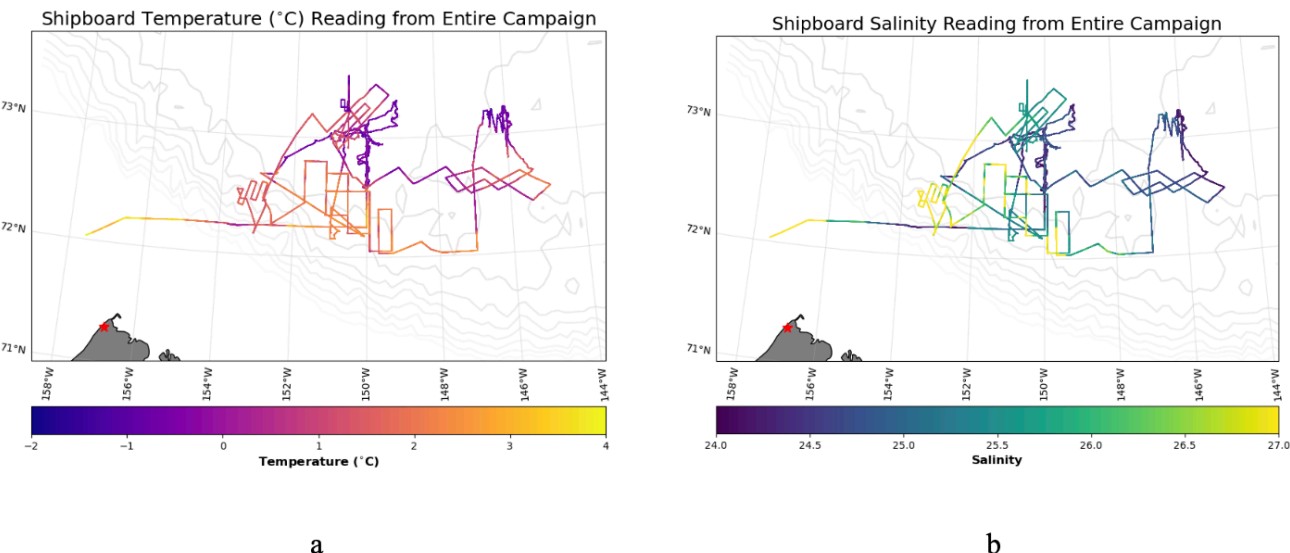

a            b

**Figure 5. (a) Temperature and (b) salinity from the shipboard TSG along the ship track from the entire SASSIE campaign.**

**2.1.2 Meteorology and Air-Sea Fluxes (Menezes and Zippel, 2023)**

Shipboard atmospheric measurements (wind speed and direction, shortwave and longwave radiation, barometric pressure, air humidity, and air temperature) were made almost continuously throughout the SASSIE cruise from two masts (on the port and



starboard side of the bow, which were instrumented with the WHOI DCFS (Direct Covariance Flux System) system, Metek uSonic-3, and Licor 7500; Figure 6) and from a barometer and radiometers mounted above the ship's wheelhouse (Table 2).

These measurements are used to calculate air-sea heat and momentum fluxes using both the direct covariance method (Edson et al., 1998) and the traditional bulk flux algorithm from state variables. This dual strategy was chosen because the presence of sea ice modifies the surface roughness and surface temperature, making the efficacy of traditional bulk flux algorithms within 5 km of ice cover uncertain (Fairall and Markson, 1987).

| Sensor | Variables measured | Sensor location and height above water level | Notes |
|---|---|---|---|
| DCFS system: Gill R3 + Systron and Donner IMU | Wind speed, wind direction, sonic temperature, wind stress, sonic temperature flux, xyz acceleration, xyz gyro rates, | Port bow; 7.9 m | Sporadic freezing of sensor head |
| Metek uSonic-3 omni | Wind speed, wind direction, sonic temperature, wind stress, sonic temperature flux | Starboard bow; 7.9 m | |
| Licor 7500 | Moisture, air temperature, air pressure | Port bow; 7.0 (sensor head) | Sporadic freezing of sensor head |
| Vaisala HMP60 | Relative humidity (RH), air temperature | Wheelhouse 10 m | Both port and starboard RH sensors failed. |
| Setra 278 Barometer | Atmospheric pressure | Wheelhouse; 10 m | |
| Kipp and Zonen CMP21 pyronometer | Downwelling shortwave radiation | Wheelhouse; 10 m | |
| Kipp and Zonen CGR4 pyrgeometer | Downwelling longwave radiation | Wheelhouse; 10 m | |

**Table 2. Meteorological sensors deployed during SASSIE. Ship position and navigation data and sea surface temperature from the Salinity Snake were also used to compute bulk flux estimates.**



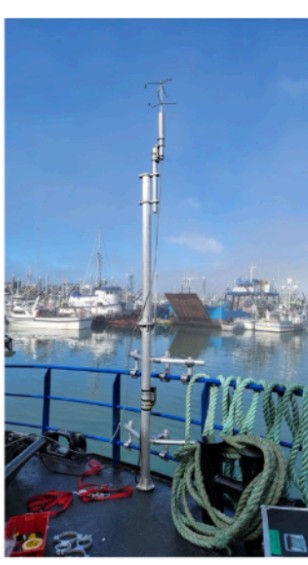
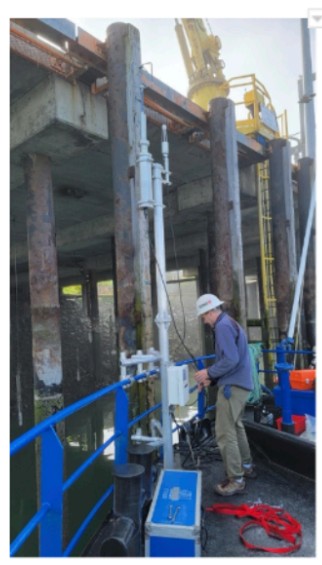

a                                         b

**Figure 6. Starboard bow air-sea flux mast instrumented with Metek (a); port bow mast instrumented with DCFS system and Licor**
**(b).**

Detailed quality control processes for the meteorological measurements are as follows:

*Wind speeds:* Mean wind speeds between the two anemometers differed by as much as 10% due to flow distortion around the
ship's superstructure. An ad-hoc correction was applied to each mean wind estimate informed by previous ship and buoy
computational fluid dynamics studies to reconcile the 10% mean wind speed errors as a function of ship-relative direction.
True wind speeds and directions are then estimated by subtracting the ship's velocity vector (estimated from the navigation
data). The two sonic anemometer time series are de-spiked, and averaged to form a blended wind speed product.

*Air temperature:* The Vaisala data reported intermittently and had numerous spikes. The Licor temperature was recorded from
a sensor in the instrument's electronics box, which was not radiation shielded and so exhibited high bias under strong shortwave
radiative forcing. The Metek sonic temperature was stable due to the heating of the sensor, but exhibited a general bias and
responded to changes in the speed of sound due to moisture. The blended temperature product combines data from these three
sensors. Figure 7 shows wind speed and direction measurements, colored by air temperature.






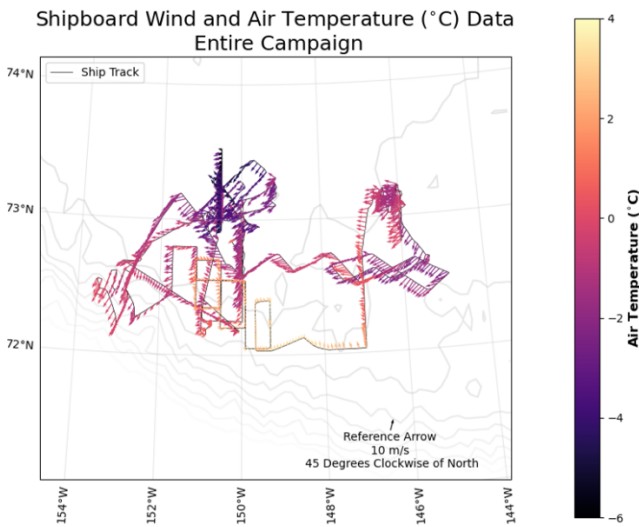

**Figure 7. Wind speed and direction measured from the RV *Woldstad* throughout the SASSIE campaign. The arrows are colored by air temperature. Bathymetry contours are shown for reference.**

*Direct momentum fluxes:* Direct covariance estimates of the wind stress are made following the Edson et al. (1998) method. Raw wind speed data are corrected using the acceleration/gyro system to remove the contamination from ship pitch, heave, and tilt motions.

   *Inertial dissipation fluxes:* Estimates of total kinetic energy (TKE) dissipation rate are made by fitting a theoretical model to
the measured power spectra of vertical velocity from the sonic anemometers. A TKE balance is assumed, and inverted for wind stress as described in Fairall et al. (1990) and Edson et al. (1991).

   *Bulk fluxes:* Bulk fluxes are estimated from the COARE 3.5 algorithm (Edson et al., 2013) using blended wind speed, air temperature products, radiometer measurements, and SST (see Section 2.1.1). Due to the failure of the Vaisala RH sensors and
the noise of the Licor 7500, bulk fluxes are run with a constant RH (RH = 89) informed by previous cruises (Persson et al. 2018). Errors in the latent heat flux are reported as the range of fluxes estimated from the range of RH values reported in Persson et al. (2018) (RH= 70 to RH=100). Figure 8 provides a summary of surface flux measurements throughout the campaign.

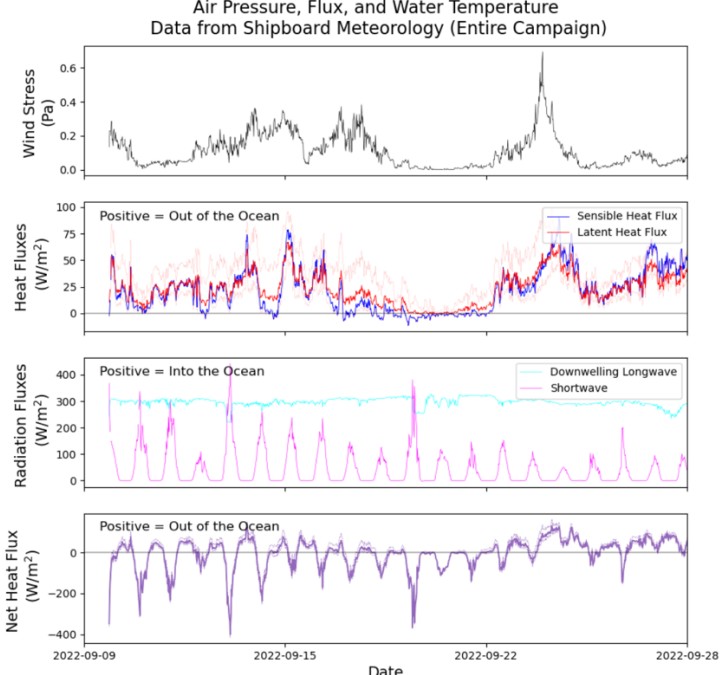

**Figure 8. Surface flux measurements measured from RV *Woldstad*. a) Wind stress (Pa). b) Latent (red) and sensible (blue) heat flux (W/m²), with dashed lines showing uncertainty of latent heat from RH values. c) Net longwave (blue) and shortwave (pink) radiation (W/m²). d) Net heat flux (solid line) and its uncertainty (dashed lines).**

Winds in the SASSIE domain were predominantly easterly (Figure 7). SASSIE captured three periods of relatively strong winds (~14-15 September, 17-18 September, 24 September) that drove turbulent cooling, and a period of very weak winds (19-20 September) during which sensible and latent fluxes were near zero. Throughout the campaign, downwelling radiation remained fairly consistent. As the daily peak in solar radiation decreased during the transition to autumn, net heat flux transitioned from predominantly warming the ocean in early September to cooling the ocean starting around 22 September (Figure 8).

**2.1.3 Salinity snake and FDOM/CDOM (Schanze, 2023)**

To measure the very near surface salinity without contamination from the ship's wake, a salinity snake (Ho and Schanze, 2020) was deployed on the RV *Woldstad* during SASSIE The salinity snake instrument consists of four main components: a boom, an intake hose, a pump, and a shipboard apparatus. A boom with a halyard is deployed to the starboard side of the vessel, which is used to position a flexible steel-reinforced rubber hose outside the wake of the ship. Water is pumped from the surface of the ocean at approximately 0.05 m depth, passes through a de-bubbler system and enters a Seabird SBE-45 TSG (note that this is completely independent from the TSG that is permanently installed on the RV *Woldstad*). Water then passes through a Turner Designs C-FLUOR fluorometer with RS232 output. The fluorometer yields measurements of fluorescent dissolved



organic matter (FDOM), which is a proxy measurement of colored dissolved organic matter (CDOM). The FDOM probe is factory calibrated and needs no further calibration. A SBE56 temperature logger inside the seawater end of the intake hose makes a continuous measurement of water temperature at ~0.05 m depth.


Ancillary data generated by the salinity snake system include the flow rate, pump suction, and instrument pressure. These values are used in the quality control of the salinity data to ensure a flow rate between 0.5 - 3 l min⁻¹ was present at the instrument.

The salinity snake collected data throughout the SASSIE cruise (Figure 9) and captured the same general temperature and salinity patterns as the shipboard TSG (Figure 4b).  FDOM patterns (Figure 10) are distinct from those of temperature or salinity.

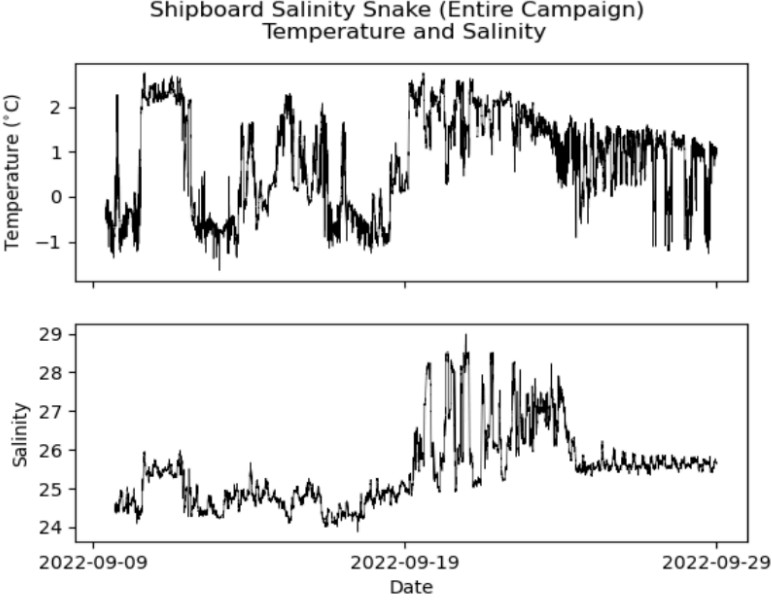

**Figure 9. Shipboard salinity snake temperature and salinity measurements throughout the entire SASSIE campaign.**



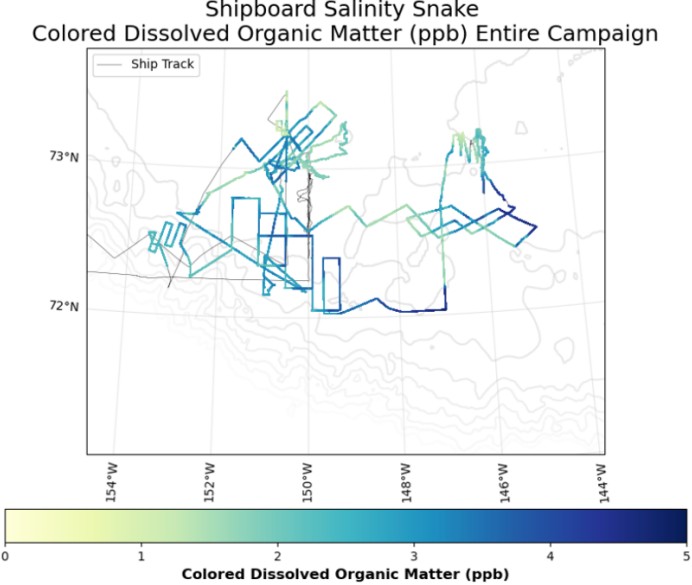


**Figure 10. Colored dissolved organic matter measurements from the shipboard salinity snake throughout the SASSIE campaign. Ship track (black line) and bathymetry contours from 1000m to 6000m shown for reference.**

### 2.1.4 ADCP (Gaube, 2023)

A hull-mounted 300 kHz Teledyne Workhorse ADCP was used to estimate vector current profiles from ~ 5 m to ~70 - 120 m

depth depending on the concentration of scattering material in the water. Unfortunately, this ship's motion could not be correctly removed from the current vectors, thus vector ocean velocity estimates could not be made. Estimates of the vertical ocean current shear were produced and are archived in the data collection, but are not displayed here.

### 2.1.5 uCTD and cCTD Measurements (Schmidgall, 2023; Pérez 2023)

The underway CTD (uCTD) system was used to measure vertical profiles of temperature and conductivity (and thus salinity

and density) at survey speed throughout the SASSIE experiment. The uCTD, a platform that was developed and built at APL-UW, was primarily deployed in open water, whereas in areas with sea ice a SonTek Castaway CTD (cCTD) was more typically used. A total of 2,518 CTD casts were collected in the Beaufort Sea (Figure 11), 2,246 of which were made with the APL uCTD.





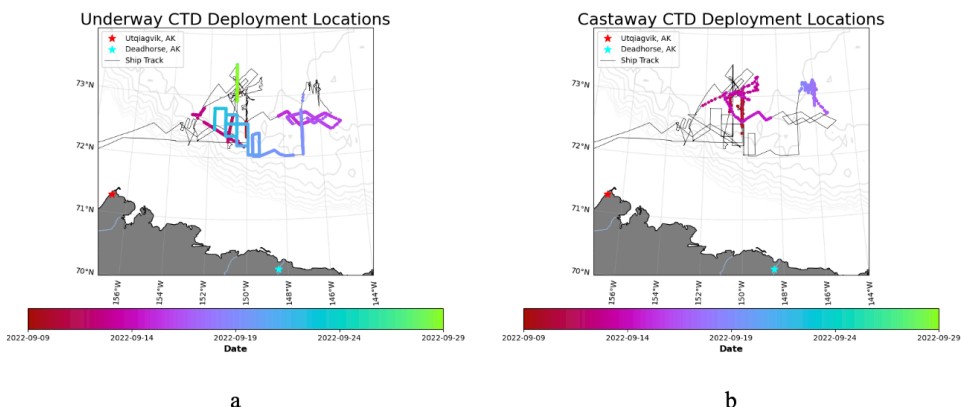

a                                        b


**Figure 11. Locations of uCTD (a) and cCTD (b) profiles colored by date, along with the ship track (black line). Individual casts are hard to distinguish as they are close together. Bathymetry data (gray lines) added for reference.**

The uCTD consists of an armored and faired RBR concerto CTD sensor (RBR, Ottawa, Canada) that was lowered and raised using an electric fishing reel, which was mounted to a stand at the aft starboard side of the RV *Woldstad* (Figure 12a). The line

was extended several meters to the aft starboard side with a boom to sample away from the ship's prop wash. Nonetheless, it must be assumed that the upper ~4 m of the ocean behind the ship were well mixed; those data must be used with caution. The RBR concerto recorded measurements of conductivity, temperature and pressure at 32 Hz. The average fall rate of the sensor was 2.1 m s$^{-1}$, giving a vertical sampling interval of ~6.5 cm. The horizontal spacing between continuous casts averaged ~0.8 km and the average depth of the casts was 100 m.

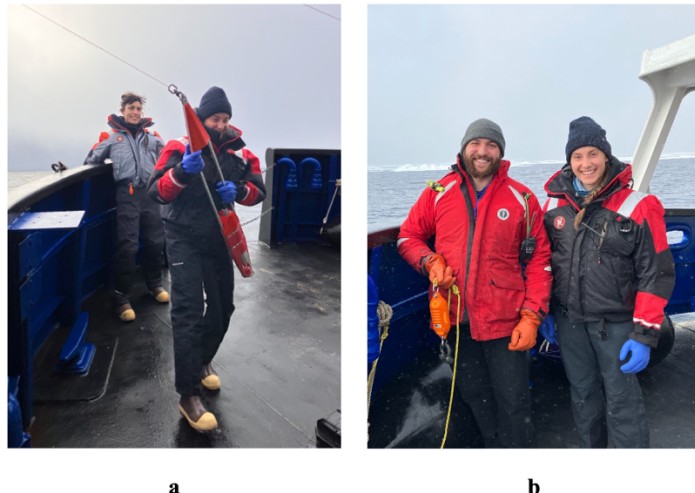

a                                        b


**Figure 12. a)** *uCTD and* **b)** *cCTD instruments*. **Photo credit Alex de Klerk (a) and Kyla Drushka (b).**

The uCTD data were offloaded via the RBR Ruskin software, quality controlled, and gridded onto a uniform vertical grid with spacing of 0.1 m. Only the data from the down casts are retained for analysis. The uCTD data were collocated with data from



the shipboard navigation system to provide position information for each cast. The uCTD was deployed continuously (in "tow-
yo" mode) as the ship steamed at 1-4 m s⁻¹ whenever conditions permitted: typically, in areas with sufficiently low ice cover
that the ship could maneuver between floes.  In slightly heavier ice cover (10-30% sea ice concentration), when continuous
profiling was not safe, the ship typically stopped every 30 minutes and a single uCTD cast was collected. During rough seas
(wind speeds greater than around 10 m s⁻¹), the uCTD could not be deployed.

In cases where the uCTD could not be used, profiles were instead collected with a cCTD – typically every 30 minutes as the
ship was underway. A total of 272 total cCTD casts were collected (Figure 12b). The cCTD is a small, rugged, and technically
advanced CTD designed for profiling to depths of up to 100 m (Figure 12b). It records temperature, conductivity, and pressure
at 5 Hz (resulting in ~0.3 m vertical resolution), as well as GPS position information for each cast. Casts reached an average
depth of 45 m, with considerable variance depending on the ship's speed and amount of line used (Figure 13,14). The accuracy
of the cCTD salinity measurements is roughly 0.1. Surface data were validated against TSG measurements for quality control
assessment.

The uCTD and cCTD measurements collected during SASSIE reveal a wealth of horizontal and vertical upper ocean structure
(Figures 13,14,15), including signatures of Pacific summer water (temperature maxima around 50 m depth), near-surface
temperature maxima typical of the region (Jackson et al., 2010), and strong lateral salinity gradients in the upper 20 m.

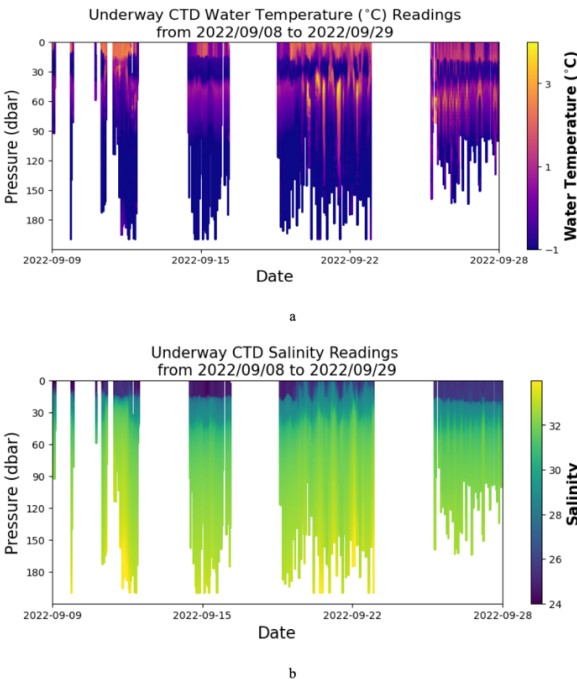





**Figure 13. (a) temperature and (b) salinity profiles measured by the uCTD.**

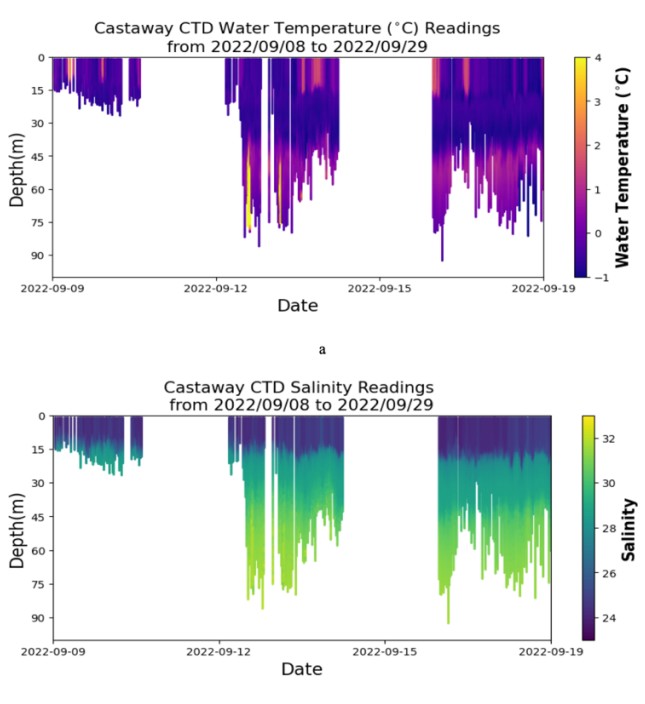

**Figure 14. (a) temperature and (b) salinity profiles measured with the Castaway CTD.**

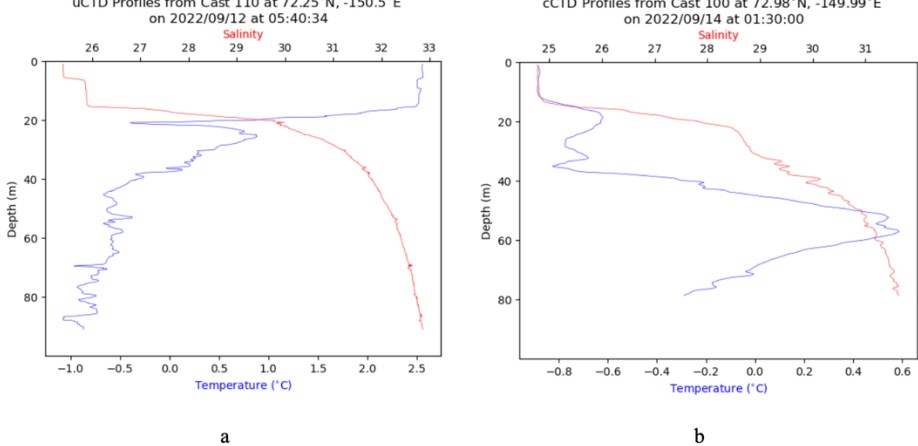

**Figure 15. (a) An example uCTD temperature (blue) and salinity (red) from 12 September at 05:40Z at 72.25N, -150.5E (b) An example cCTD cast from 14 September at 01:30Z at 72.98N, -149.99E. Note the subsurface temperature maxima at ~20 m and at ~55 m in (b). The uCTD cast was taken in open water, whereas the cCTD cast was taken within the ice.**



### 2.1.6 S-band Radar (Drushka, 2023b)

The RV *Woldstad* is equipped with a Furuno 2137S-BB S-band marine navigation radar. The ship relied heavily on this radar for navigating through sea ice, particularly during the night and when visibility was low due to fog. X-band radar data have been used in ship-based studies of sea ice (e.g., Lund et al., 2018); S-band radar data have also been used for this purpose (Haykin et al., 1985).

During the SASSIE cruise, screenshots of the ship's S-band radar data were collected by splitting the signal and turning it into a video feed using a screen capture device (Epiphan AV.io HD, Figure 16). This video feed was acquired using a command-line video conversion program called "ffmpeg", which saved screenshots as jpegs at a specified time interval: typically, every 60 seconds while the ship was in and around the ice. During JetSSP deployments in the ice, images were acquired every 10 seconds to map the ice evolution in higher resolution. While the ship was in open water for longer than one day, S-band acquisition was paused.

The S-band radar was actively used by the ship's captain and crew for navigation, and the radar settings were adjusted frequently, leading to inconsistencies in the data. Typically, the ship set the radar range to 6 to 12 nm (11 to 22 km) during the daytime and 0.75 to 3 nm (1.4 to 6 km) during the night or in heavy ice concentrations. The captain and mate adjusted the range, heading orientation, and gain as needed to reduce clutter and identify safe routes through the ice. During the night, the radar was set to a lower-brightness night-mode. The radar was switched to standby mode when personnel were on top of the pilot house.



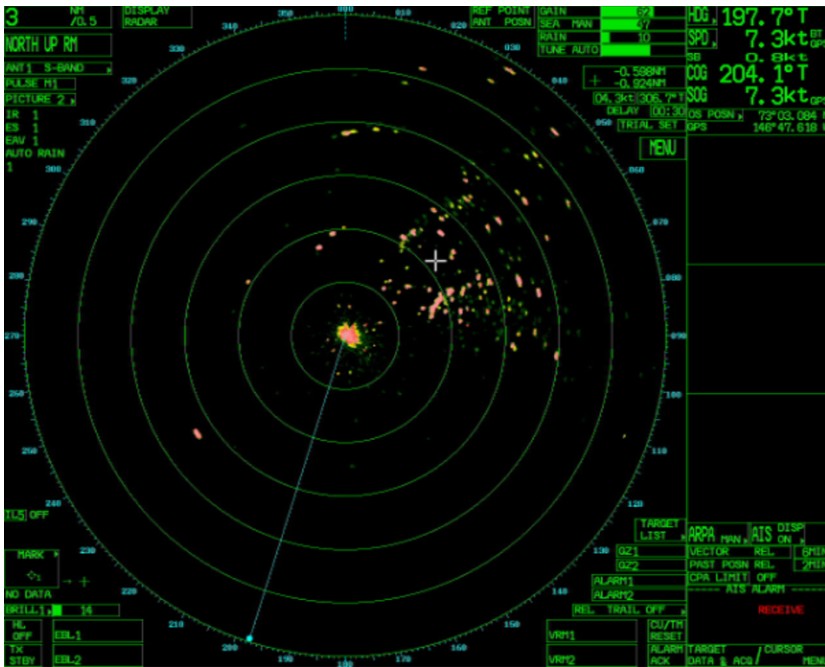


**Figure 16. S-band radar snapshot on 19 September 2022 at 00:34Z as the ship was transiting southward out of the ice. Metadata is shown on the image: total range is 3 nm with 0.5 nm per range ring (upper-left of image); north is up in the image; image is oriented such that up corresponds to north; ship's heading is shown as the cyan line on the image and the text in the upper-right corner. The presence of ice is indicated by the red-yellow patches in the upper-right quadrant of the range rings. The bright area in the center of the image is sea clutter.**


Metadata were extracted from the images using MATLAB's computer vision toolbox. The pixels outside of the largest range ring were stripped from each image, and pixels within the range rings were georectified and stored as GeoTIFF images.

### 2.1.7 Bottle samples for δ18O (Drushka, 2023c)

Water samples were collected periodically throughout the cruise for $\delta^{18}O$ isotope analysis to help differentiate the sources of

the freshwater signals in the SASSIE domain - presumed to be a combination of sea ice meltwater, meteoric water (likely from the Mackenzie River), and Pacific seawater. A total of 45 water samples were collected, including two duplicates (Figure 17). Water samples were collected either from a GoFlo bottle lowered from a line over the side of the ship to approximately 1 m (N=7 samples), 5 m (N=12), or 30 m (N=1) depth. An RBR duet temperature-pressure logger was attached to the GoFlo bottle, providing accurate measurements of the depth and water temperature of the sample. An additional 21 water samples were

collected from the Salinity Snake, which pumped water from ~0.05 m depth. Finally, samples were collected during two ice stations on 13 September: tailings from drilling into the sea ice were collected, melted, and stored in sample bottles.

All samples were stored in 20 ml glass scintillation bottles, sealed with parafilm, and later analyzed using a mass spectrometer for $\delta^{18}O$ isotopes at the Oregon State University Stable Isotope Laboratory.



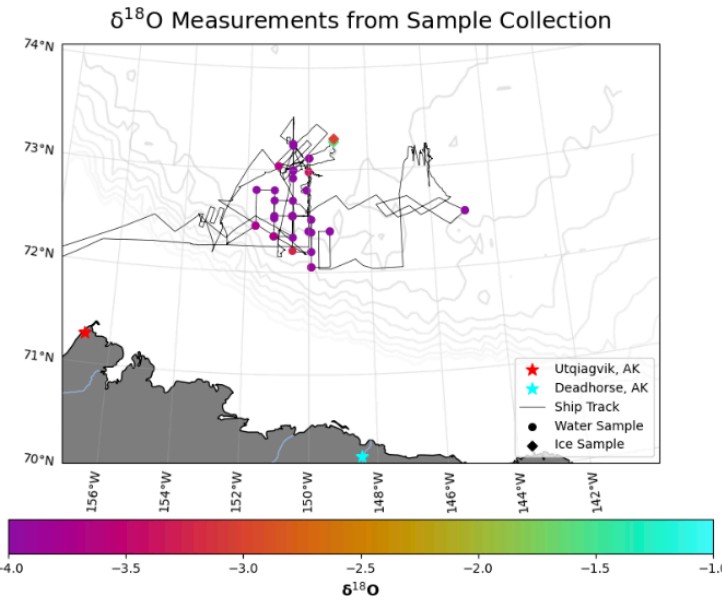

**Figure 17. Locations and measurements of the δ¹⁸O ratio measurements and the value recorded. Water samples are shown as circles (·) and ice samples are shown as diamonds (◆). The track of the RV *Woldstad* is shown as a black line.**

## 2.2 Piloted & Drifting Platforms

### 2.2.1 Wave Gliders (Thomson, 2023a)

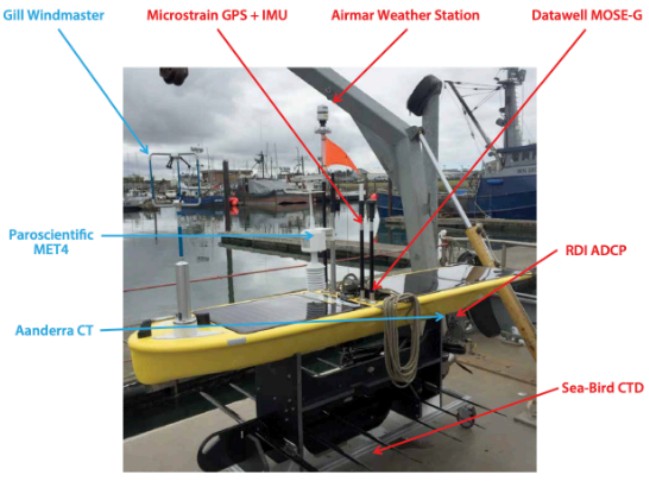


**Figure 18. Labeled photo (Thomson et al., 2017) showing the instruments on Wave Glider SV3-153. Photo credit Jim Thomson.**





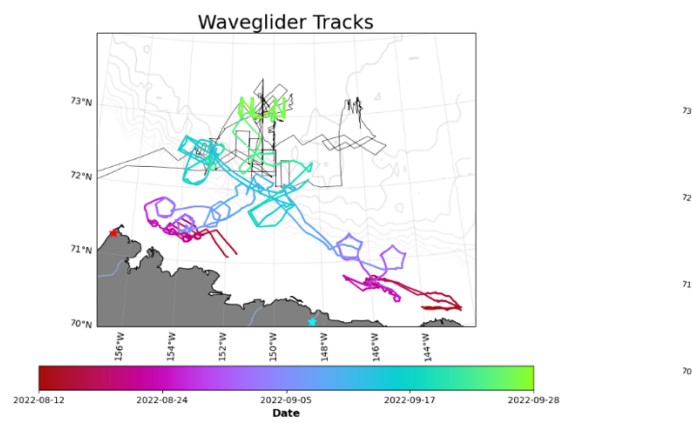

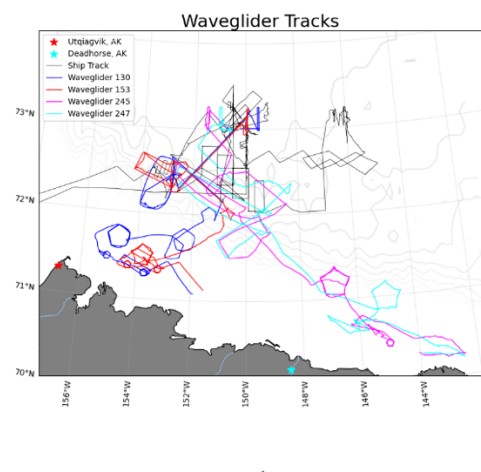

| a | b |

**Figure 19. Paths of the four Wave Gliders throughout the course of the SASSIE campaign, colored by time (a) and Wave Glider number (b). Track of the RV *Woldstad* (black line) and bathymetry contours (gray lines) are also shown for reference.**

Four Wave Gliders (model SV3 built by Liquid Robotics, Inc.) from UW/APL were operated throughout the SASSIE campaign. The Wave Gliders were deployed from the RV *Ukpik* operating out of Prudhoe Bay in early August. They were recovered for battery charging and redeployed twice from the RV *Woldstad* in September (Table 3), and the final recovery was in late September from the *Woldstad*. Each Wave Glider operated for approximately 50 days and covered almost 1000 nm, often within 10 nm of the sea ice.

| Wave Glider (SV3) number | Deployment location | Deployment date | Recovery Location | Recovery Date |
|---|---|---|---|---|
| 130 | 71.08 °N, 151.91 °W | 14-Aug-2022 20:00:00 | 72.66 °N, 152.86°W | 12-Sep-2022 18:30:00 |
| | 72.66 °N, 152.86 °W | 12-Sep-2022 18:00:00 | 73.08 °N, 150.19 °W | 25-Sep-2022 02:30:00 |
| | 73.08 °N, 150.19 °W | 25-Sep-2022 02:00:00 | 73.00 °N, 150.19 °W | 28-Sep-2022 19:30:00 |
| 153 | 71.12°N, 151.54 °W | 14-Aug-2022 20:00:00 | 72.63°N, 152.80°W | 12-Sep-2022 18:30:00 |
| | 72.63°N, 152.80 °W | 12-Sep-2022 18:00:00 | 73.06 °N, 150.19 °W | 25-Sep-2022 02:30:00 |
| | 73.06°N, 150.19 °W | 25-Sep-2022 02:00:00 | 73.00 °N, 150.42°W | 28-Sep-2022 19:30:00 |
| 245 | 70.31°N, 144.04 °W | 12-Aug-2022 23:00:00 | 72.58 °N, 152.66°W | 12-Sep-2022 18:30:00 |
| | 72.59°N, 152.66 °W | 12-Sep-2022 18:00:00 | 72.52 °N, 150.01°W | 21-Sep-2022 18:30:00 |
| | 72.51°N, 150.03 °W | 21-Sep-2022 18:30:00 | 73.00 °N, 150.77°W | 28-Sep-2022 19:30:00 |
| 247 | 70.30°N, 143.89 °W | 12-Aug-2022 23:00:00 | 72.55 °N, 152.59°W | 12-Sep-2022 18:30:00 |
| | 72.55°N, 152.59 °W | 12-Sep-2022 18:00:00 | 72.99 °N, 150.53°W | 26-Sep-2022 01:30:00 |
| | 73.00°N, 150.53 °W | 26-Sep-2022 01:00:00 | 72.99 °N, 150.62°W | 28-Sep-2022 19:30:00 |

**Table 3. Deployments of Wave Gliders throughout the SASSIE campaign. Times in UTC.**



Each Wave Glider included a standard package to measure meteorological data (Airmar WX200 sensor), waves (GPSWaves by Thomson et al, 2018), ocean currents (RDI 300 KHz ADCP), near-surface temperature and salinity (Aanderaa CT) and sub-surface temperature and salinity (Seabird GPCTD) (Table 4). Some of these sensors, such as the ADCP and the Airmar, only operated for a fraction of the mission as a battery saving strategy. Salinity measurements from the Aanderaa CT and the Seabird GPCTD sensors agree with a root-mean-square difference of 0.1 (not shown). Two of the wave gliders include profiling winches for CTD measurements beneath the sub (8 to 150 m). These prototype systems collected a total of 39 profiles before failing; that data is included in the L1 data, but did not pass quality control for inclusion in the L2 data.

| Sensor | Variables Collected | Notes |
|---|---|---|
| Airmar WX200 sensor | Air temperature and pressure, wind speed | Only operated for a fraction of the deployment time to conserve battery life. |
| GPS waves | Wave height, period, and direction | Also, frequency spectra and directional moments |
| RDI 300 KHz Acoustic Doppler Current Profiler | Ocean current speed and direction | Only operated for a fraction of the deployment time to conserve battery life. |
| Aanderaa model 4319B CT sensor | Conductivity and temperature at 0.2 m and 1 m depth | Some gaps in data due to occasional sensor failure. |
| Seabird GPCTD | Conductivity, temperature and depth at 4 or 8 m | Only on SV3-130 (4 m) and SV3-153 (8 m) |
| D2 profiling CTD | Conductivity, temperature, depth up to 150 m per cast (about once per day while operational). | Only equipped on Wave Gliders SV3-245 (31 casts total) and SV3-247 (8 casts total) |

**Table 4. List of sensors and variables collected by Wave Gliders.**



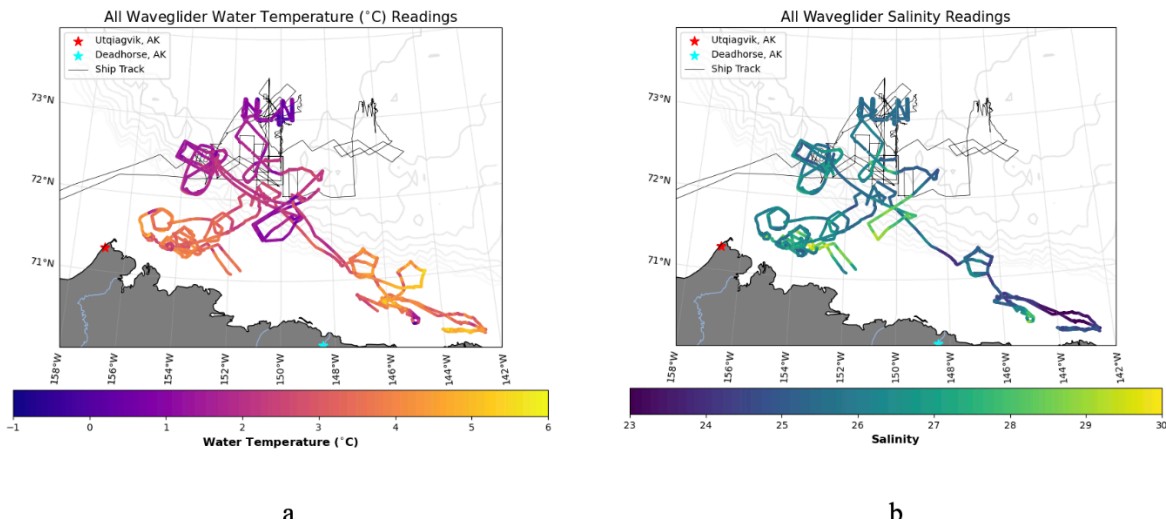

**Figure 20. Paths of all four Wave Gliders throughout the course of the SASSIE campaign, colored by salinity (a) and water temperature (b) at 0.25 m depth, except in the case of salinity measurements of Wave Glider 153, which were taken at 1 m depth. Track of the *RV Woldstad* (black line) and bathymetry contours (gray lines) are also shown for reference.**

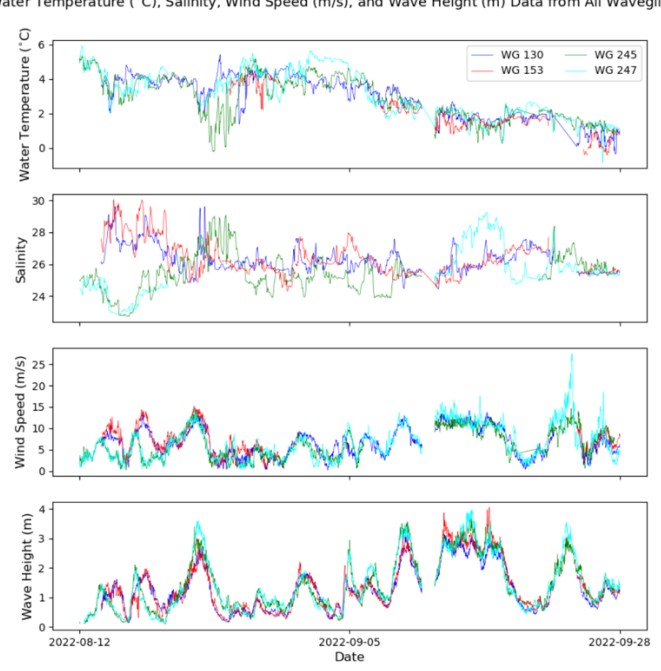

**Figure 21. Time series of water temperature, salinity, wind speed, and wave height measurements made by the four Wave Gliders throughout the SASSIE campaign. Temperature and salinity measurements are from 0.25 m depth, except for the salinity measurements from Wave Glider 153 that were made at 1 m depth.**



### 2.2.2 JetSSP (Drushka, 2023d)


The JetSSP is an autonomous USV designed to measure salinity gradients in the upper meter of the ocean. The system consists of a 3.1 m long gasoline powered jet-kayak that is controlled using an onboard suite of drone hardware to enable autonomous navigation (Figure 22). Telemetry data is broadcast over a 900 MHz antenna as well as independent AIS and Iridium trackers. The vehicle has fuel capacity to perform 12+ hr autonomous sampling missions traveling at speeds of 4 to 5 kts. A 1 m long keel is bolted to the underside of the vehicle and outfitted with three SBE 49 CTDs that sampled below the water line (Figure 22b).


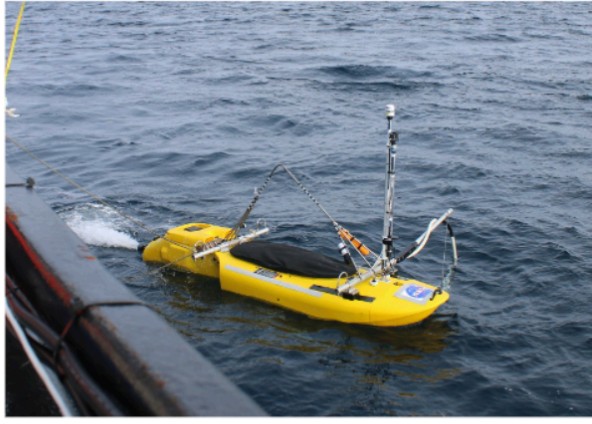
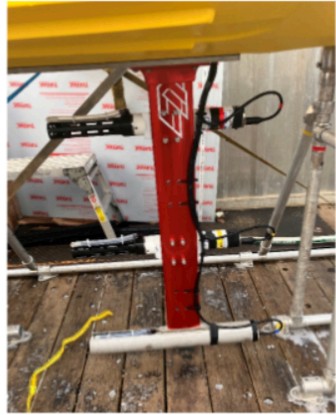

a                                                                    b

**Figure 22. a) The JetSSP just after deployment. Note the forward meteorology mast and the hose off the port side for the salinity snake. b) The 1 m-long keel of the JetSSP with three CTDs; the front end of the platform is to the left in the photo. Photo credit Alex de Klerk**


 In addition, a miniaturized "salinity snake" consisting of a SBE 45 TSG, a positive displacement pump, and a vortex debubbler is mounted internally in order to sample water from a hose dragged at the surface. Using these four instruments JetSSP is able to continuously measure salinity and temperature at nominal depths of 0.05, 0.25, 0.65 and 1.05 m, capturing the very near-surface signature of horizontal and vertical salinity and temperature gradients associated with freshwater layers (Figures 23 and 24). A forward mast with an Airmar 200WX-IPX7 weather station reports air temperature and pressure, and GPS-corrected true wind speed and direction. Data are logged at 6 Hz with a WETLabs DH-4 logger.






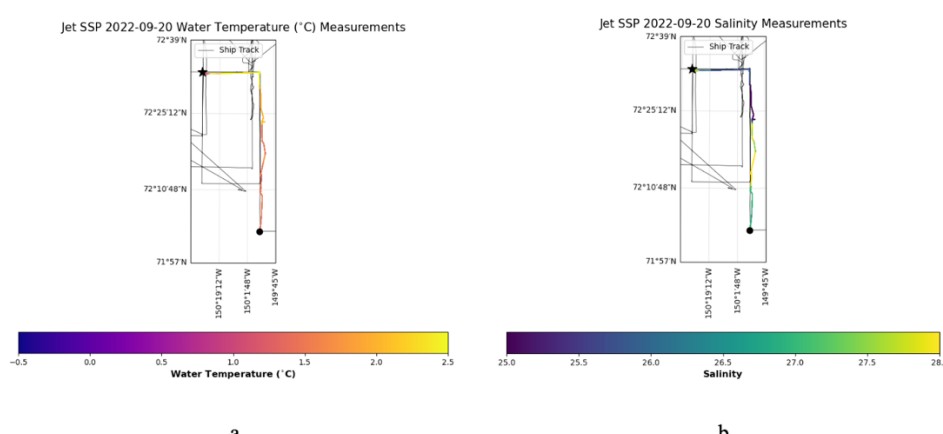

a                                              b

**Figure 23. Tracks of JetSSP deployment 4 on 09-20-2022 color-coded by temperature (a) and salinity (b). The thin lines show the track of the RV *Woldstad*. Black circle marks initial deployment location and Black star marks the recovery.**

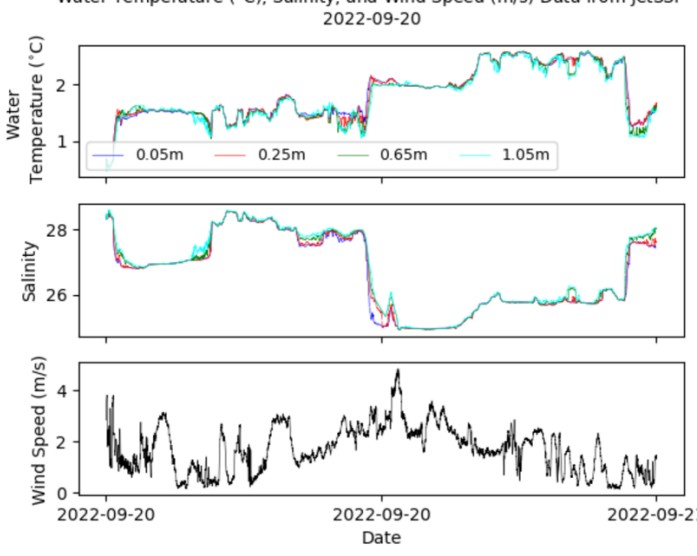


**Figure 24. Sea water temperature (top), salinity (middle), and wind speed (bottom) records from Jet-SSP deployment on 20 September.**

The JetSSP platform was deployed from the ship and successfully collected data seven times (Table 5, Figure 25). On the eighth and final deployment, the platform was lost.




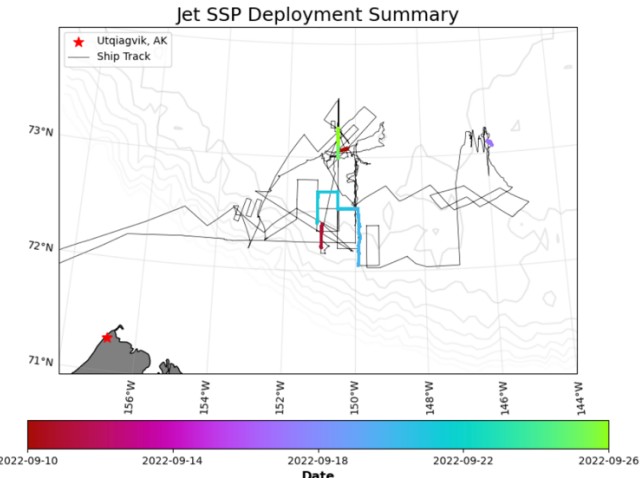

**Figure 25. Paths of Jet SSP deployments throughout the course of the SASSIE campaign, colored by time. Track of the RV *Woldstad* (black line) and bathymetry contours (gray lines) are also shown for reference.**

| Deployment number | Deployment date | Deployment Location | Deployment duration (hours) |
|---|---|---|---|
| 1 | 10 September | 73.07°N, 150.25°W | 2.8 |
| 2 | 11 September | 72.20°N ,150.95°W | 3 |
| 3 | 17 September | 73.06°N 145.84°W | 3.5 |
| 4 | 20 September | 72.05°N 149.90°W | 9.3 |
| 5 | 21 September | 72.55°N 150.13°W | 10.5 |
| 6 | 25 September | 73.01°N 150.50°W | 10 |
| 7 | 26 September | 73.11°N 150.50°W | 3.8 |

**Table 5. Deployment dates, locations and durations for the Jet-SSP. Dates are local time.**

### 2.2.3 SWIFT (Thomson, 2023b)

Five SWIFTs (Thomson 2012) were repeatedly deployed and recovered in the Beaufort Sea during the SASSIE field campaign (Figure 26). SWIFTs are freely drifting buoys that measure waves, winds, temperature, salinity, turbulence, and velocity profiles in a surface-following reference frame (Figure 27).



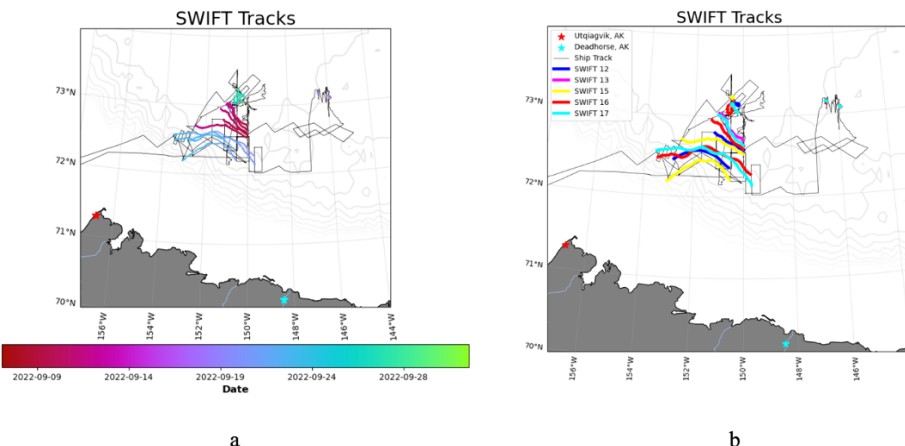


**Figure 26. Paths of the five SWIFTs over multiple deployments throughout the course of the SASSIE campaign, colored by time (a) and SWIFT number (b). Track of the RV *Woldstad* (black line) and bathymetry contours (gray lines) are also shown for reference.**

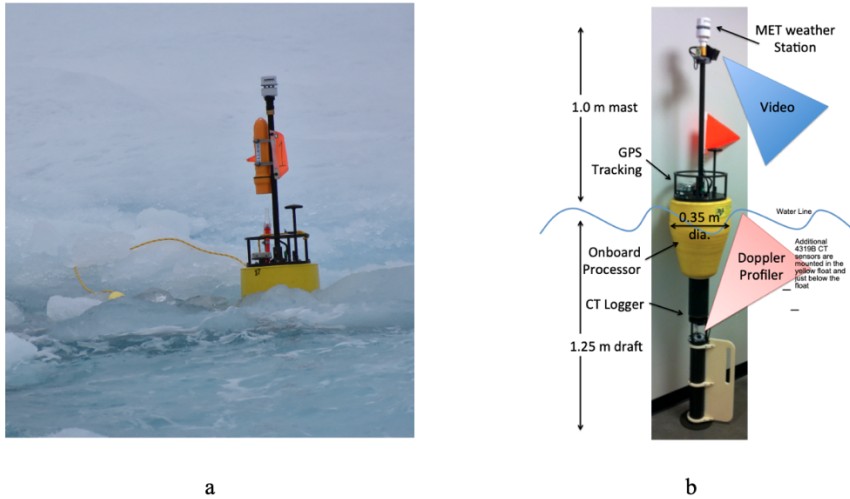

**Figure 27. a) A SWIFT floating within slushy ice. Photo credit E J Rainville. b) A SWIFT in the lab with dimensions and notation.**

For SASSIE, additional conductivity and temperature (CT) sensors (Aanderaa model 4319B) were added to most buoys (Figure 28b). The configurations varied from a full suite of three CT sensors at depths of 0.2, 0.6, and 1.2 m on some buoys, to a single CT sensor at 0.6 m on other buoys. These configurations varied between deployments, as repairs and replacements required. Buoys without three CTs included a camera with images of surface conditions (including sea ice, Figure 28a). The reported salinity values are determined onboard each CT sensor using a factory calibration. The uppermost CT (at 0.2 m depth) on each



SWIFT has a salinity offset of almost 1 that is caused by proximity of the inductive core to the buoy floatation collar. During high sea states, the uppermost CT is also contaminated by bubbles.

The deployments were organized into a series of plays, coordinated with other drifting buoys, floats, and shipboard sampling by the RV *Woldstad*. The SWIFT collections and conditions are summarized Table 6. Data collection during Play 1 employed five bursts of raw data per hour. Data collection during all other plays used a single burst (512 s) of raw data per hour, to conserve battery power.  The L2 data are statistical products for each burst that are calculated from the raw data (collected at 4 Hz).


During several plays, a line of SWIFTs was deployed across a lateral surface salinity gradient to capture the evolution of the gradient over time and space (e.g., Figure 28).

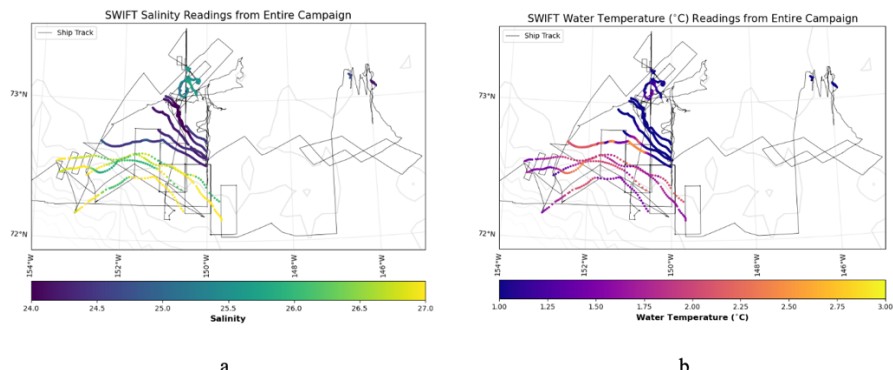

Figure 28.  Salinity (a) and temperature (b) measurements taken by SWIFTs throughout the SASSIE campaign. Track of the RV
*Woldstad* (black lines) and bathymetry data (gray lines) shown for reference.

| SWIFT identifier | Play number | Deployment location | Deployment date |
|---|---|---|---|
| 12 | 1 | 72.5860°N, 149.9649°W | 09-Sep-2022 21:24:00 |
|  | 4 | 72.3682°N, 150.5954°W | 22-Sep-2022 10:00:42 |
|  | 5 | 73.1734°N, 150.4843°W | 25-Sep-2022 22:04:29 |
| 13 | 1 | 72.7033°N, 149.9960°W | 10-Sep-2022 03:36:00 |
| 15 | 1 | 72.5380°N, 149.9941°W | 09-Sep-2022 22:12:00 |
|  | 4 | 72.2162°N, 150.5449°W | 21-Sep-2022 10:10:16 |
|  | 5 | 73.2440°N, 150.4968°W | 28-Sep-2022 00:10:55 |
| 16 | 1 | 72.5973°N, 149.9980°W | 10-Sep-2022 02:10:00 |



| | 3a | 73.0772°N, 145.8576°W | 17-Sep-2022 17:44:17 |
|---|---|---|---|
| | 3b | 73.1378°N, 146.4767°W | 18-Sep-2022 18:44:05 |
| | 4 | 72.2864°N, 149.7039°W | 20-Sep-2022 06:44:35 |
| | 5 | 73.0961°N, 150.4869°W | 25-Sep-2022 23:10:42 |
| 17 | 1 | 72.6435°N, 149.9938°W | 10-Sep-2022 02:48:00 |
| | 3a | 73.0777°N, 145.8670°W | 17-Sep-2022 18:10:20 |
| | 3b | 73.1567°N, 146.4397°W | 18-Sep-2022 18:12:06 |
| | 4 | 72.1495°N, 149.6523°W | 20-Sep-2022 04:13:30 |
| | 5 | 73.1458°N, 150.4996°W | 26-Sep-2022 20:13:38 |

**Table 6. A list of deployment dates and locations for SWIFT buoys.**

### 2.2.4 HydroBuoys (Steele, 2023)

The SASSIE HydroBuoys were designed to measure both surface hydrography (i.e., SST and SSS) as well as subsurface
temperature and salinity to provide thermal and haline stratification. They were deployed during the September 2022 SASSIE
cruise and not recovered, thus providing persistent observations through the fall and even into the winter.

These buoys were based on surface-drifting buoys deployed in the Arctic Seas by the UpTempO project
(http://psc.apl.washington.edu/UpTempO/; Castro et al., 2016, Banzon et al., 2020), which generally focuses on temperature
observations only. For SASSIE, conductivity sensors were added to measure salinity. A variety of sensor configurations were
used on a total of 11 buoys deployed during SASSIE, from surface-only temperature and salinity to a heavily instrumented
string of sensors extending down to 60 m depth (Figure 29; Table 7). All buoys provide data at 10 minute intervals. The buoys
generally drift with the surface currents (or ice in the winter); several were equipped with a drogue.



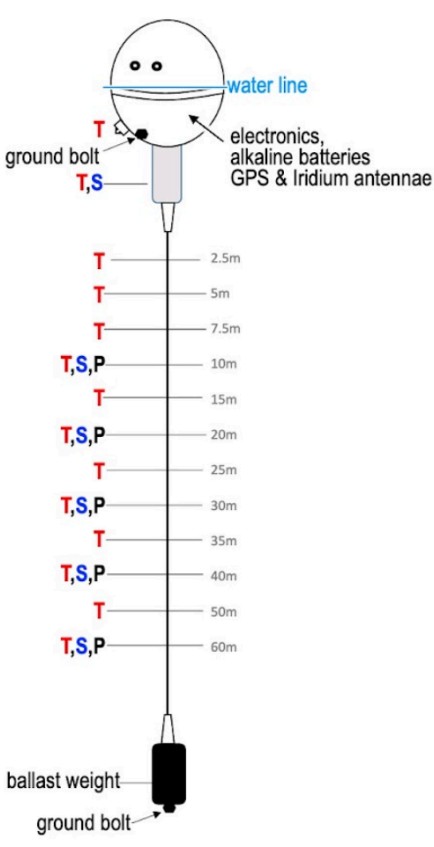


**Figure 29. Schematic of SASSIE HydroBuoy 009, with sensors measuring ocean temperature (T), conductivity from which salinity is derived (S), and pressure (P). Other buoys had as many or fewer sensors (Table 7). This buoy did not include a drogue, whereas some others did.**

| Drifter number | Deployment Location | Deployment Time | Temperature Data Depths (m) | Salinity Data Depths (m) | Pressure Data Depths (m) |
|---|---|---|---|---|---|
| HydroBuoy 001 | 72.25 °N, 151.37°W | 09September 2022 00:00:00 | 0.14, 0.44, 5 | 0.38, 5 | 5 |
| HydroBuoy 002 | 72.25 °N 151.38 °W | 09September 2022 00:00:00 | 0.14, 0.44 | 0.38 | N/A |
| HydroBuoy 003 | 72.40 °N 150.02 °W | 10September 2022 00:00:00 | 0.25, 2.5, 5, 7.5, 10, 15, 20, 25, 30, 35, 40, 50, 60 | 10, 20, 30, 40, 60 | 10, 20, 30, 40, 60 |
| HydroBuoy 004 | 73.07 °N 150.26 °W | 11September 2022 00:59:03 | 0.14, 0.44, 5 | 0.38, 5 | 5 |





| | | | | | |
|---|---|---|---|---|---|
| HydroBuoy 005 | 73.26 °N 149.13 °W | 13September 2022 16:41:15 | 0.25, 2.5, 5, 7.5, 10, 15, 20, 25, 30, 35, 40, 50, 60 | 10, 20, 30, 40, 60 | 10, 20, 30, 40, 60 |
| HydroBuoy 006 | 72.61 °N 145.35 °W | 15September 2022 20:00:56 | 0.14, 0.44 | 0.38 | N/A |
| HydroBuoy 007 | 72.84 °N 146.00 °W | 17September 2022 14:00:56 | 0.14, 0.44 | 0.38 | N/A |
| HydroBuoy 008 | 72.03 °N 149.25 °W | 20September 2022 00:00:00 | 0.14, 5 | 5 | 5 |
| HydroBuoy 009 | 72.03 °N 149.26 °W | 20September 2022 00:00:00 | 0.25, 0.57, 2.5, 5, 7.5, 10, 15, 20, 25, 30, 35, 40, 50, 60 | 0.51, 10, 20, 30, 40, 60 | 10, 20, 40, 60 |
| HydroBuoy 010 | 73.03 °N 150.82 °W | 25September 2022 15:30:56 | 0.14, 5 | 5 | 5 |
| HydroBuoy 011 | 73.03 °N 150.82 °W | 25September 2022 15:30:56 | 0.14, 5 | 5 | 5 |

**Table 7 Deployment location, time, and nominal depths (i.e., when the sensor string is hanging vertically) of available temperature, salinity, and pressure data of SASSIE HydroBuoys.**

Buoys with subsurface sensor strings also had ocean pressure sensors to determine changes in hydrography sensor depths owing to several reasons, i.e., (1) sensor string uplift owing to wind or ice pushing the surface hull, which makes the trailing sensor string rise up in the water column (this is very common), (2) sea ice ridging, which pulls up the sensor string into the ice (this often happens during winter), and (3) dragging on the ocean bottom (this generally happens when a buoy transits over the continental shelf, which did not happen during SASSIE). All buoys generally drifted west, northwest, and north after deployment (Figure 30). All buoys performed well until they encountered ice (generally in late September through early December 2022), at which time communications stopped for many but not all buoys. As of September 2023, 4 buoys were still reporting, all of which had drifted northward to ~ 86N, 135W over the Alpha Ridge. Four buoys were still reporting sea surface temperature (SST), three were reporting sea surface salinity (SSS), and one was reporting surface and subsurface temperature and salinity.





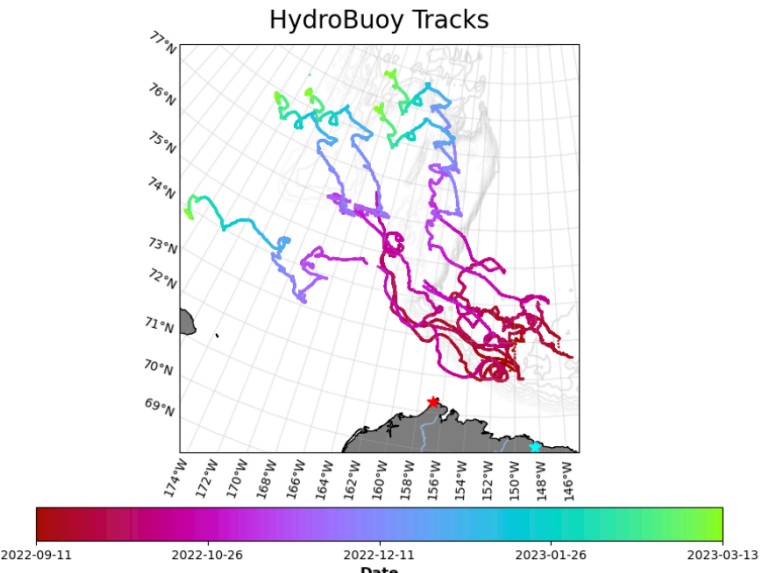

**Figure 30. Paths of the SASSIE HydroBuoys throughout the SASSIE campaign and beyond. Shown are drift tracks from deployment in September, 2022 through March, 2023.**

An example of buoy data is provided in Figure 31. The ocean temperature data shows gradual cooling at the surface, and warm

intrusions at depth that are likely summer Pacific Water entering the southern Beaufort Sea from the north-eastern Chukchi

Sea. The ocean salinity data shows declining stratification in the upper water column as fresh meltwater is mixed downward

and salinified via ice growth and brine rejection.

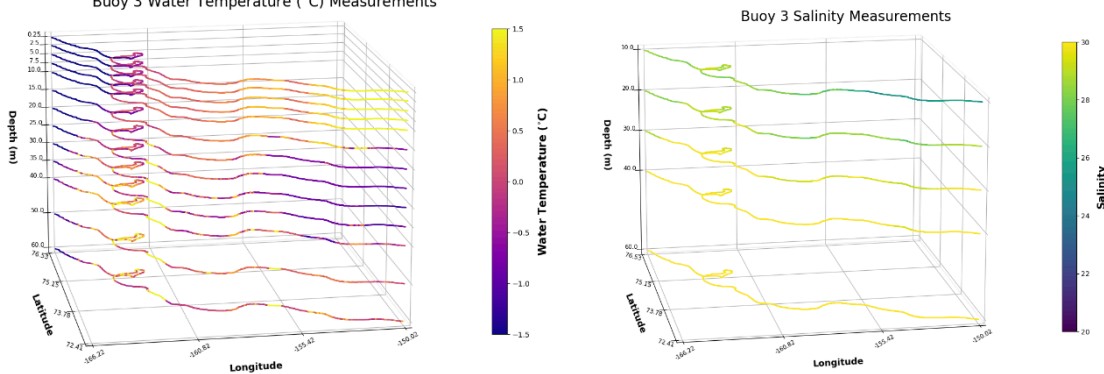

**Figure 31. Water temperature and salinity measurements by HydroBuoy 3.**



### 2.2.5 Under-Ice Float (Shcherbina, 2023)

A UIF is a heavily-instrumented autonomous profiling vehicle based on the Mixed-layer Lagrangian float design (e.g., D'Asaro, 2003). The SASSIE UIF was outfitted with two pumped Seabird CTDs mounted on the top and the bottom of the float, a narrow-beam upward-looking sonar, and an upward-looking camera for ice imaging (Figure 32).

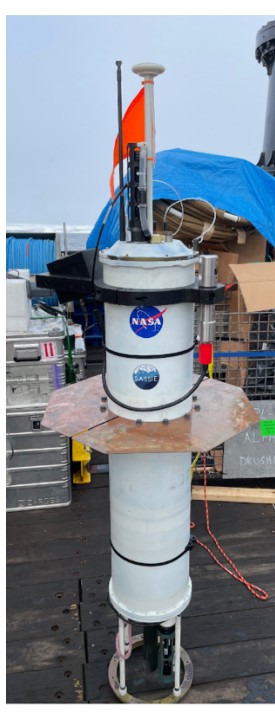


**Figure 32. The UIF on the deck of the Woldstad before deployment. The float is about 2 m tall from the deck to the top of the antennas. Note CTDs at top and bottom of the float, upward-looking camera mounted near the top of the float body on the left, and an ice-profiling sonar attached to the right-hand side. Photo credit Jim Thomson.**

The UIF was deployed from RV *Woldstad* on 10 September 2022 at 20:05 UTC at 73.12°N, 149.93°W within the marginal

ice zone (Figure 33). During its deployment, the float made frequent vertical profiles, surfacing when possible (i.e., when the sonar detected open water) to telemeter data. The last data transmission occurred on 23 October 2022.



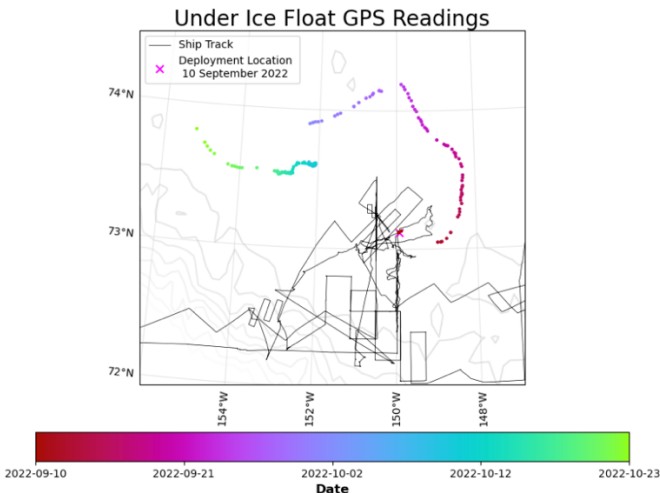

**Figure 33. GPS Readings from the Under Ice Float colored according to date. The gaps in the record indicate the periods of heavy ice that prevented float surfacing. Black line denotes ship track.**

UIF profiling was focusing on the upper 40-50m of the water column to resolve the evolution of the cold and fresh mixed layer in the later stages of summer ice melt and the transition to freeze-up (Figure 34). Particular attention was paid to the upper meters under the ice or open water, where vertical CTD resolution down to 1 cm was achieved through reduced profiling speed and increased sampling rate (Figure 35). Periodically, the UIF spent time hovering 10-15 meters below the surface, utilizing its sonar and camera to survey the ice topography and capture photos (Figure 36).


The UIF CTD records (Figures 34-35) show general increase in salinity, decrease in temperature, and deepening of the mixed layer in September-October. These changes were due to entrainment of saltier pycnocline water accompanied by surface heat loss. Advective changes in the water mass structure were also evident below 30m (within the pycnocline). Until September 30, mixed layer temperature stayed >0.1°C above the freezing point. At that time, the ice cover remained fragmented and

consisted of old brash ice (Figure 36a). Even though the temperature approached freezing by October, new ice formation was intermittent. Calm weather periods allowed periodic formation of pancake (Figure 36a) and nilas ice. These new ice formation events were interspersed with prolonged periods of open water and rough seas. Ice formation intensified and started to show signs of consolidation in the second half of October.





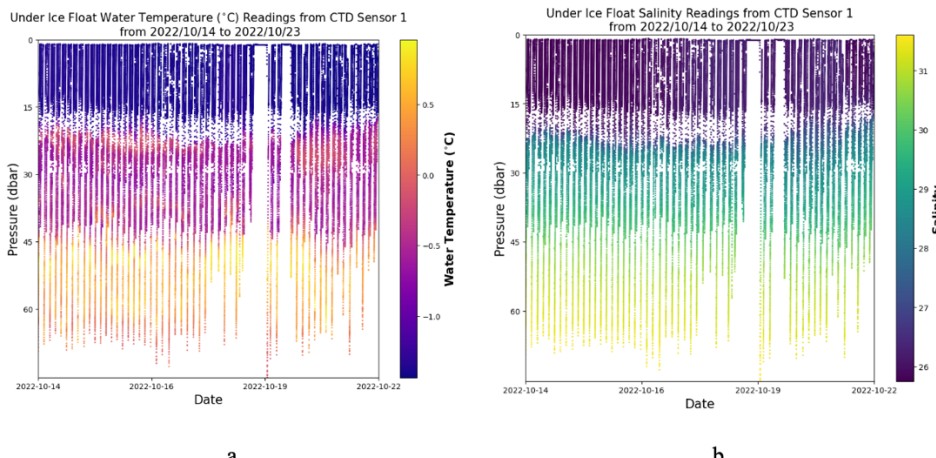

**Figure 34. 2D scatterplots of water temperature (a) and salinity (b) readings from UIF CTD sensor 1 (at the bottom of the UIF; Figure 32) throughout its deployment.**

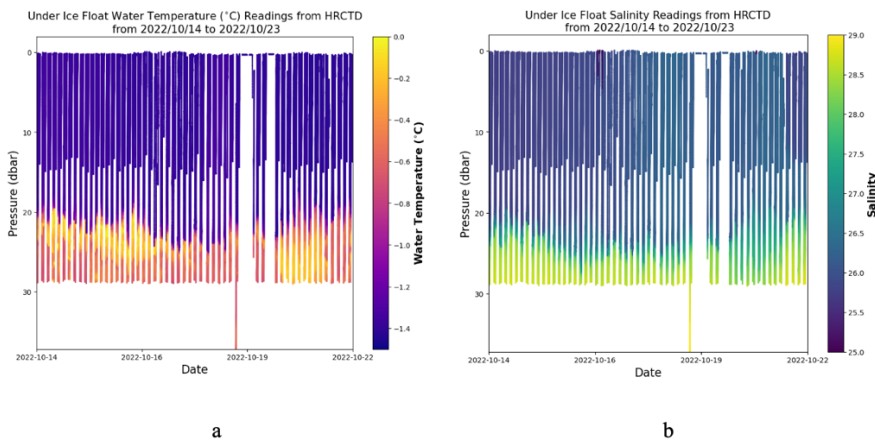

**Figure 35. 2D scatterplots of water temperature (a) and salinity (b) readings from the UIF high resolution CTD sensor throughout its deployment. The high resolution CTD is seen attached to the top of the float in Figure 32. Note the different color scale and vertical axes between this and the previous figure.**



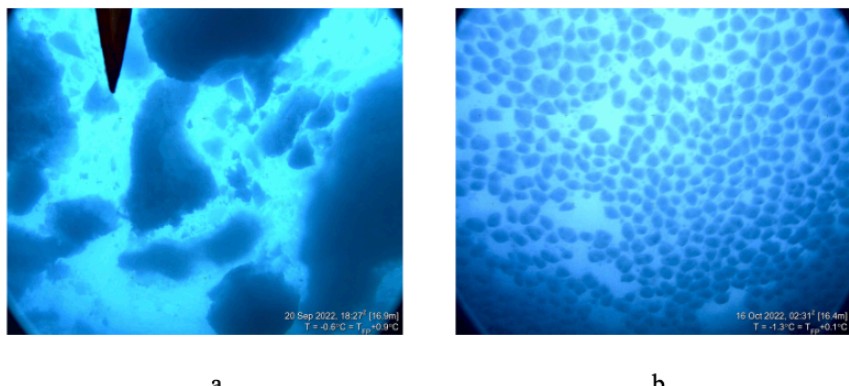

a                                                      b

**Figure 36. Examples of UIF ice imagery. a) Broken old ice (brash) commonly observed during the first half of the mission. b) New pancake ice formation observed periodically during the freeze-up. Images are annotated in the lower-right corner with the date, time, UIF depth at that time, observed temperature and its departure from the freezing point ($T_{FP}$). The black shape visible in (a) is the UIF flag.**

### 2.2.7 ALTO and ALAMO Floats (Jayne and Steele, 2023)

Seven profiling floats (Table 8) manufactured by MRV systems were deployed during SASSIE. These floats provide autonomous profiles of ocean temperature and salinity and were set to sample over the upper 100 m. They thus provide information about SST, SSS, and thermal and haline stratification. The floats profile vertically by changing their buoyancy, and drift laterally with the local currents. Since the floats were sampling rapidly (multiple profiles per day) over the upper 100 m, their drift is a complex integration of upper ocean current speed and direction.

| Float number | Deployment location | Deployment date | Final transmission of 2022 |
|---|---|---|---|
| 11131 - ALTO | 72.25 N, 153.04 W | 8 September 2022 | 10 October |
| 11132 - ALTO | 72.25 N, 150.00 W | 8 September 2022 | 14 October |
| 11133 - ALTO | 72.25 N, 154.95 W | 8 September 2022 | 14 October |
| 11136 - ALTO | 72.25 N, 151.51 W | 8 September 2022 | 24 October |
| 9097 - ALAMO | 72.5 N, 145.0 W | 15 September 2022 | 12 October |
| 9101 - ALAMO | 72.0 N, 149.5 W | 19 September 2022 | 4 October |
| 9098 - ALAMO | 73.12 N, 150.5 W | 25 September 2022 | 12 October |

**Table 8. Deployment Dates and locations of the ALTO and ALAMO floats. Three additional ALAMO floats were deployed (two on 9 September 2022 and one on 10 September 2022) but did not collect any data.**




Four of the seven working floats were relatively larger (133 cm long, 16.5 cm, 19.2 kg) ALTO floats (Figure 37), and three were relatively smaller (91 cm long, 12.2 cm diameter, 8.4 kg) ALAMO floats. All floats stopped reporting owing to advancing ice by late September through October 2022 (Figure 38). The ALTO floats included an "ice avoidance" algorithm with the goal of surviving winter. Unfortunately, none of these floats reported in summer 2023. The ALAMO floats did not include this feature, in order to maximize data collection during freeze-up.


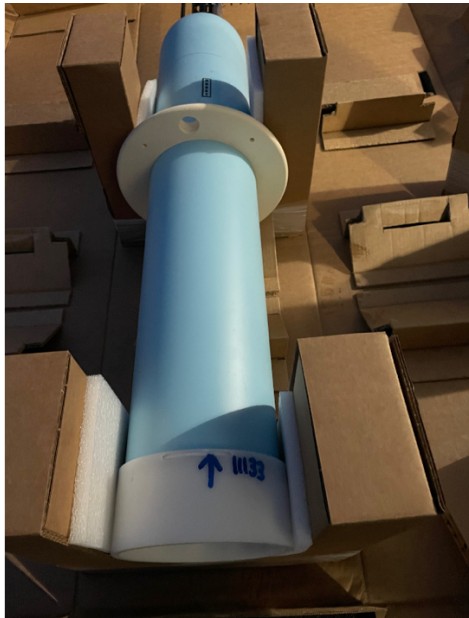

**Figure 37. An ALTO float waiting to be deployed.**

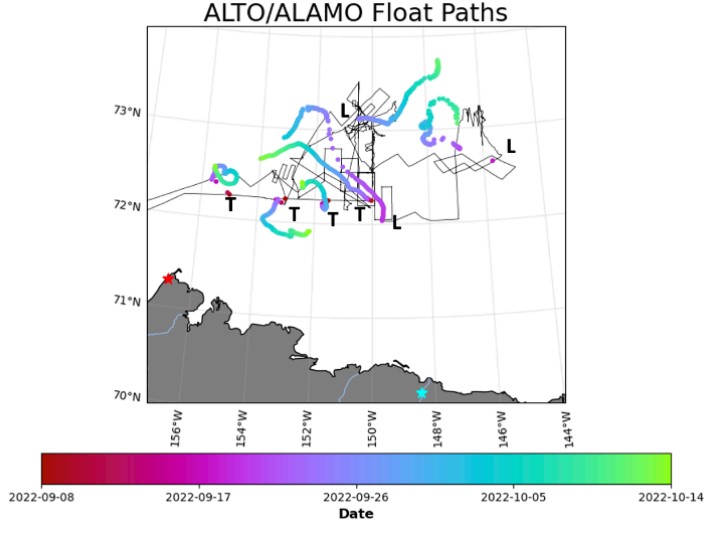

**Figure 38. Paths of the ALTO (designated "T") and ALAMO (designated "L") floats throughout the field campaign colored by date.**



An example of float data is provided in Figure 39. A clear signal of subsurface warm layers is evident, especially to the southeast at the start of the drift where warm summer Pacific Water is likely entering the Beaufort Sea from the northeastern Chukchi Sea. The data show a rather complex temperature profile that might also include warming from local radiative fluxes (i.e., a Near-Surface Temperature Maximum) which will be investigated using full hydrographic and meteorological analyses.

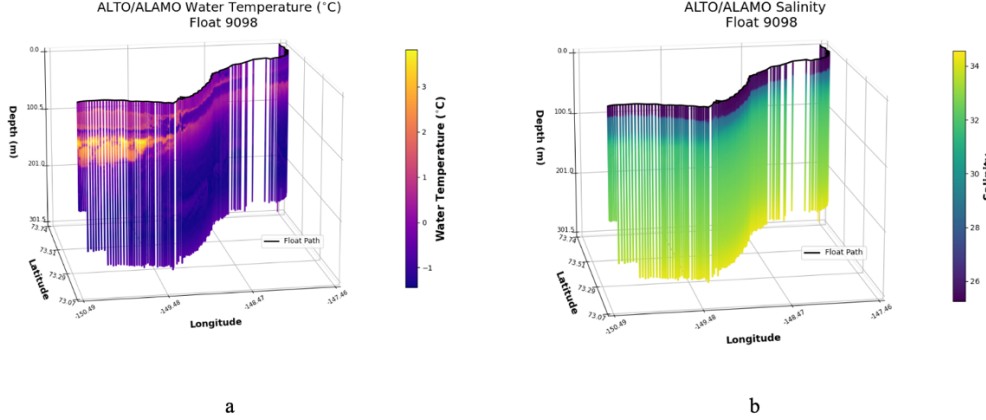

a                                        b


**Figure 39. Temperature (a) and Salinity (b) readings from ALAMO float 9098.**

## 2.3 PALS Aircraft

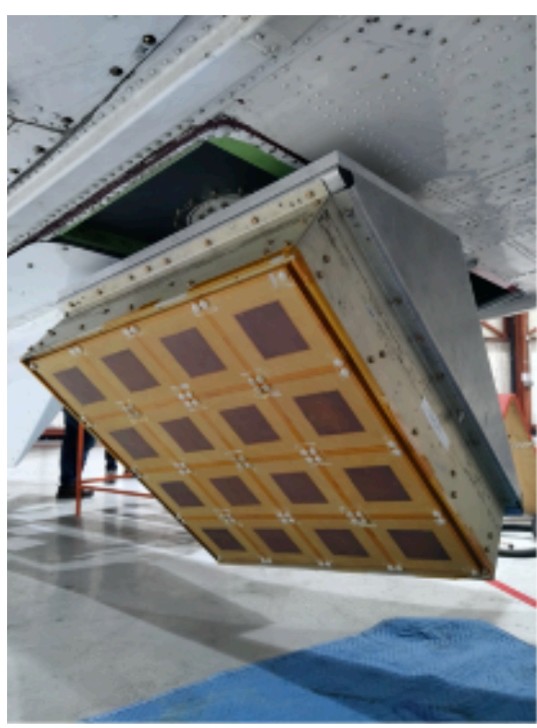

**Figure 40. PALS patch array antenna installed underneath the DC3-T Basler (BT-67) aircraft. The antenna is mechanically rotated**
**360 degrees via a scanner motor (not seen) and covered with a radome before flight operations.**



To link in situ SSS measurements to SSS patterns across broader spatial scales, airborne measurements of SSS were made with the JPL Passive/Active L-band System (PALS) microwave radiometer (Wilson et al., 2001a,b; Yueh and Chaubell, 2011). The aircraft measurements provided broad-scale context and enabled quantification of SSS and SST, their horizontal gradients, and sea ice signals. PALS measures at the same L-band frequencies as the NASA Soil Moisture Active Passive (SMAP)

satellite instrument, with significantly increased spatial resolution and accuracy over shorter time-scales. PALS measures at 1.4 GHz with 24 MHz bandwidth and has the capability of measuring at 1.2 GHz with an 80 MHz bandwidth. The combination of these two frequencies significantly increases the sensitivity to salinity in cold waters. PALS can provide 300 m to 1500 m horizontal resolution when flying at altitudes of 1 to 3 km. The airborne PALS radiometer has significantly smaller measurement noise compared to satellite radiometers because airborne platforms have far longer scene integration times and

therefore a greater reduction of uncorrelated measurement errors.

After averaging, PALS can theoretically measure SSS in cold waters (SST < 5°C) with 0.2-0.25 resolution over 1 km horizontal scales. During SASSIE, the instrument suite also included a compact wideband (750 MHz) C- and X-band radiometer to measure SST and ocean wind speed, respectively, at approximately 0.3 K and 1 m s$^{-1}$ resolution. The PALS instrument also

has an on-board combined infrared (IR) and visible imaging camera that covers a 45-degree swath. The IR camera provides radiometric temperatures for each point on 1.56 m x 1.95 m resolution per acquisition with <50 mK sensitivity when flying at an altitude of 1 km; the visible camera has 0.25 m x 0.33 m resolution when flying at an altitude of 1 km. For surveys over partial ice cover, the visible and IR imagery provide high-resolution estimates of ice concentration for comparison with and validation of SAR-derived ice concentration. The combination of the embedded GPS and IMU modules with the dual visible

and thermal camera capability provides time-stamped and geolocated radiometric measurements that can be synchronized with the PALS radiometer measurements.

PALS was installed in a Basler DC3-T aircraft operated by Kenn Borek Air Limited, a Canadian operator. Five PALS flights were made during SASSIE (Table 9): flights 1 and 3 were over the RV *Woldstad* and flights 2, 4 and 5 were over Wave Gliders

(Figure 41).

| Flight number | Date (2022) |
|---|---|
| 1 | 11 September |
| 2 | 14 September |
| 3 | 16 September |
| 4 | 17 September |





| 5 | 18 September |
|---|---|

**Table 9. Dates of PALS flights.**

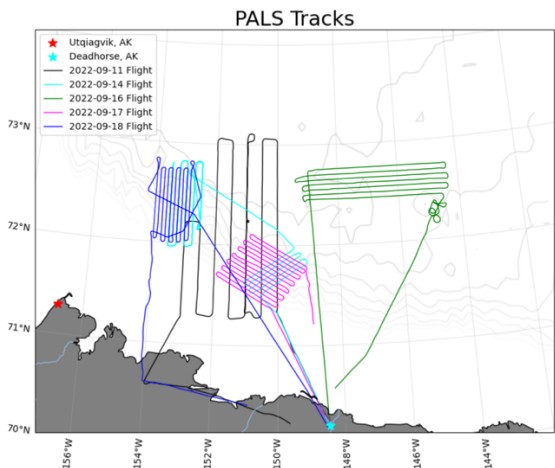

**Figure 41. Map of the PALS flights. Different colors indicate different flights (Table 9) as shown in the legend at top left. Thin gray lines show bathymetry contours.**

The PALS airborne measurement data processing consists of converting measured uncalibrated counts to polarized microwave brightness temperature values at 1.2 and 1.4 GHz, that eventually gets converted to SSS.

The initial calibration process from uncalibrated measured counts to brightness temperature (Tb) utilizes internal calibration sources present within the PALS radiometer (noise-diode and reference load) that provide instantaneous gain and offset calibration coefficients. On-board temperature sensors allow constant tracking of the instrument temperature variations as the aircraft changes altitudes and outside temperature varies. PALS calibration also takes into account any potential scan dependent variations within the data observed from sources such as the internal scan motor, or the external radome etc. Another key data

processing step is detecting and excising radio frequency interference (RFI) from the Tb measurements. During SASSIE flights, 1.2 GHz passive microwave data saw extreme RFI corruption making the data almost unusable. 1.4 GHz radiometer though was relatively clean.

Airborne Tb measurements at 1.4 GHz, see contributions from direct and reflected solar and galactic reflection, rain, RFI, SSS,

SST, and ocean wind vector or sea surface roughness. In order to derive salinity from brightness temperature other parameters need to be corrected for or removed. Additionally, at cold SST of less than 10°C, the sensitivity of brightness temperature to salinity is nearly 3 times less than tropical temperatures. SASSIE flew in rain-free conditions and took advantage of the 360-degree scan to correct for wind direction signatures and potential solar or galactic intrusions within the data. SASSIE flew an on-board IR sensor for SST, but due to mostly cloudy conditions had to rely on ancillary SST values from satellites. Wind



speed and resulting white-cap foam on the ocean surface is the most difficult contributor to correct. We use ancillary data such as ASCAT as well as on-board measurements from a C/X band radiometer to derive relative wind speed trends observed during flight lines. Tb to SSS retrieval is done using the Klein-Swift model for the Dielectric Constant of Sea Water, though other models such as Meissner-Wentz and GW-model were also tested. The absolute value in all models is slightly different at cold waters, but the relative variation of salinity during flight lines is similar. We anchor derived SSS absolute values to "ground-

truth" data obtained from the wave-glider assets.

## 3. Plays

The SASSIE cruise revolved around five major "plays" (Table 10), each of which was essentially a mini-experiment that targeted a particular region, feature, or set of conditions using a tailored sampling strategy and suite of platforms. The shipboard and aircraft teams determined the location and strategy for each play based on local conditions such as weather, sea ice,

presence of salinity or temperature gradients, as well as relevance to the SASSIE objectives.

| Play # | Description | Start date (2022) | End date (2022) |
|---|---|---|---|
| 1 | Marginal ice zone survey | 8 September | 13 September |
| 2 | Open water survey | 14 September | 16 September |
| 3 | Ice survey | 16 September | 18 September |
| 4 | Drifter-following boxes | 19 September | 23 September |
| 5 | Transects across ice-open water transition | 25 September | 29 September |

**Table 10. Summary of SASSIE plays.**

### 3.1 Play #1: Marginal ice zone survey

Play 1 began as RV *Woldstad* entered the MIZ on 8 September, deploying five ALTO/ALAMO profiling floats, five

HydroBuoys, and five SWIFTs during the following days to capture the transition between open ocean and moderate sea ice cover (Figure 42). The ship focused primarily on the sea ice covered waters north of around 72°30'N, while Wave Gliders and the aircraft captured open waters to the south. As the ship navigated through the heaviest ice cover of the expedition, the UIF was deployed and water samples were collected at two ice stations to provide end-members for δ18O isotope analysis. Key observations included horizontal surface salinity gradients of 2 to 3 from open water to ice, with strong surface salinity

variability south of the ice edge and homogeneous salinities within ice-covered waters (Figure 43).

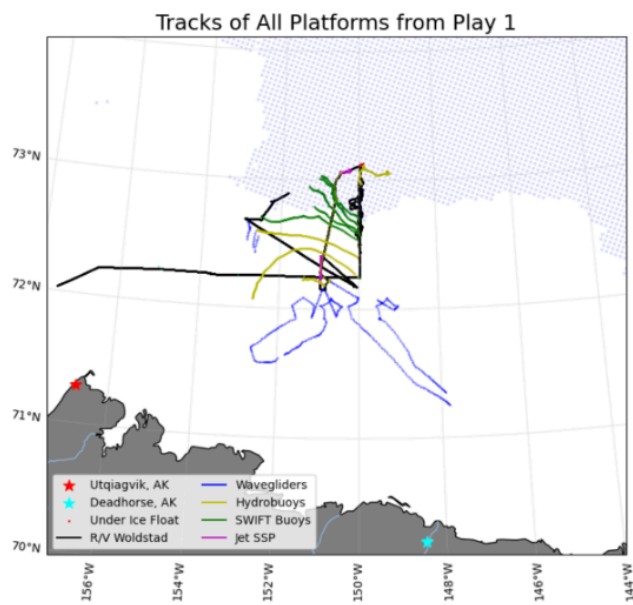

**Figure 42. A map showing the tracks of all six platforms during Play 1. Blue stippling indicates the presence of ice on 11 September 2022 as determined from the MASIE-NH ice product.**

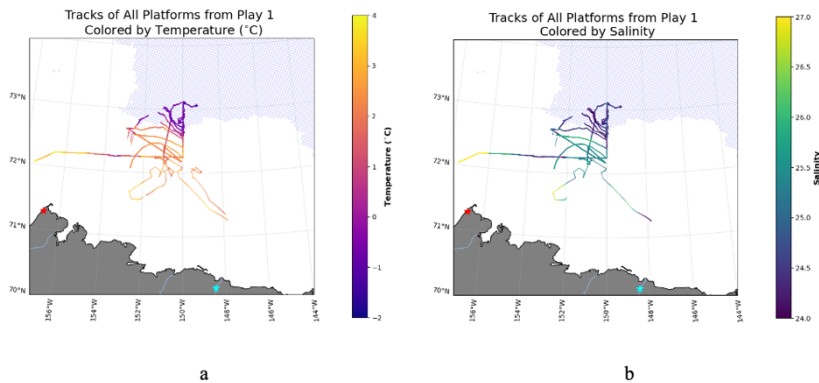

**Figure 43. Temperature (a) and salinity (b) recorded closest to the sea surface by each asset (except UIF) during Play 1. Blue stippling indicates the presence of ice on 11 September 2022, as determined from the MASIE-NH ice product.**

## 3.2 Play #2: Open water survey

During Play 2, the ship surveyed south of the ice edge between 150°W and 145°W, forming a zig-zag pattern to maximize sampling of horizontal variability (Figures 44 and 45). Winds and sea state were high, so profiling with the uCTD and JetSSP

was limited. Conditions for flying were excellent, and the aircraft performed two surveys: one day over the Wave Gliders

southwest of the ship in open water and another day in long east-west transects over the ship. The ship reached its easternmost point during this play.

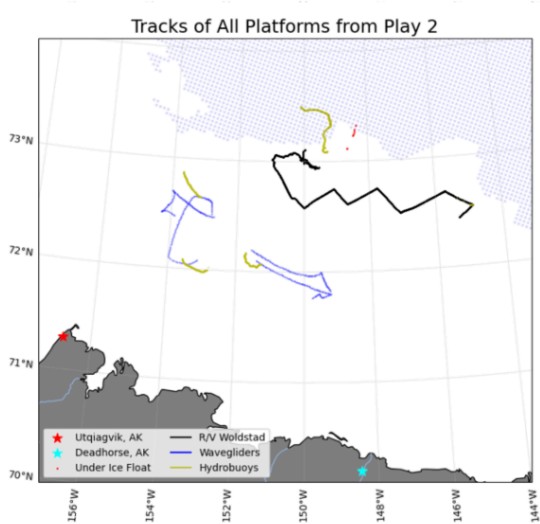

**Figure 44. A map showing the tracks of all six platforms deployed during Play 2. Blue stippling indicates the presence of ice on 15 September 2022 as determined from the MASIE-NH ice product.**

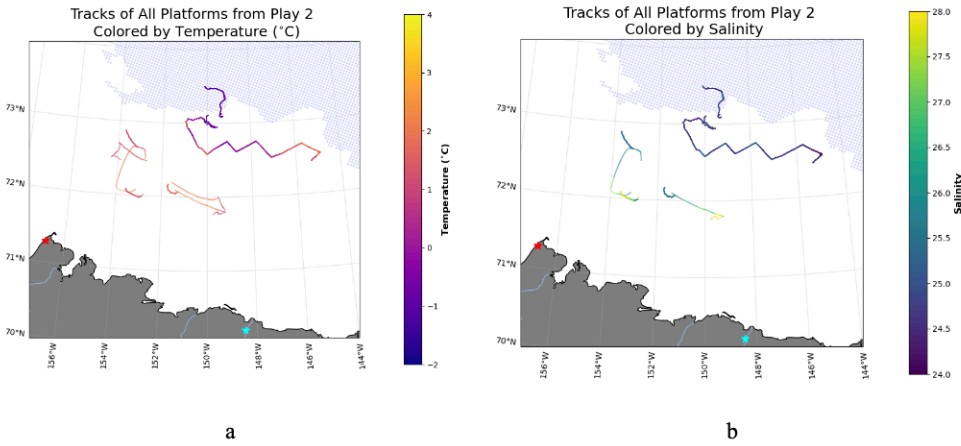

**Figure 45. Temperature (a) and Salinity (b) recorded closest to the sea surface by each asset during Play 2. The Under Ice float has been excluded. Blue stippling indicates the presence of ice on 15 September 2022, as determined from the MASIE-NH ice product.**

## 3.3 Play #3: Ice survey

On 16 September the *Woldstad* entered the sea ice to avoid heavy seas as Typhoon Merbok hit the Bering Strait (Figure 46 and 47). One HydroBuoy was deployed, and two SWIFTs and JetSSP were deployed in moderate ice cover to examine small-



scale gradients within the sea ice. Aircraft flights 3 and 4 (Table 9) over Wave Gliders to the southwest were carried out over two days.


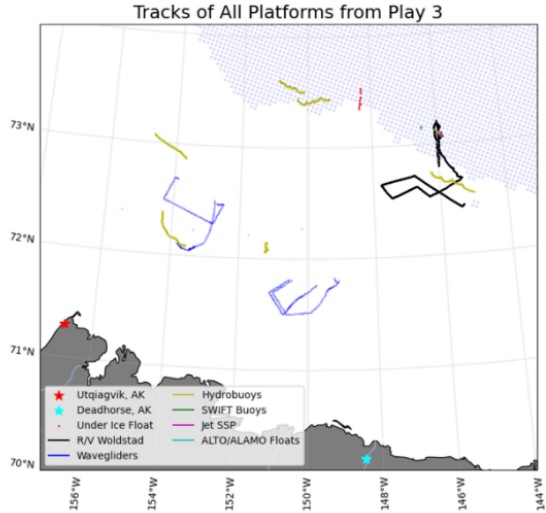

**Figure 46. A map showing the tracks of all six platforms during Play 3. Blue stippling indicates the presence of ice on 17 September 2022, as determined from the MASIE-NH ice product.**

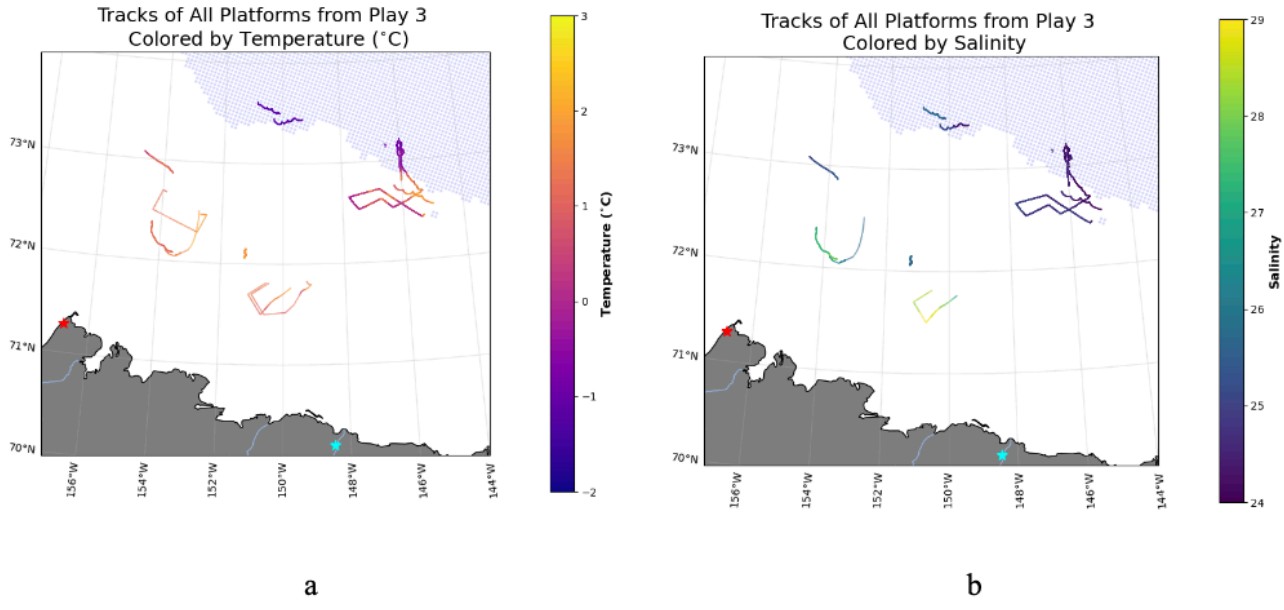

**Figure 47. Temperature (a) and salinity (b) recorded closest to the sea surface by each asset during Play 3. The Under Ice float has been excluded. Blue stippling indicates the presence of ice on 17 September 2022, as determined from the MASIE-NH ice product.**



### 3.4 Play #4: Drifter-following boxes

Play 4 focused on capturing the spatio-temporal variability of a strong surface salinity front in open water. On 19 September,
five assets were deployed in a north-south line across the gradient. As these assets drifted north-westward with the gyre
circulation on the following three days, the ship steamed in a series of overlapping boxes around them (Figure 48), collecting
continuous uCTD measurements and deploying the JetSSP next to the ship to provide measurements from the surface down
to ~100 m. Additional SWIFTs were deployed on 21 and 22 September to target signals of interest. Strong sub-mesoscale
fronts were observed, with significant variability in both space and time (Figure 49).

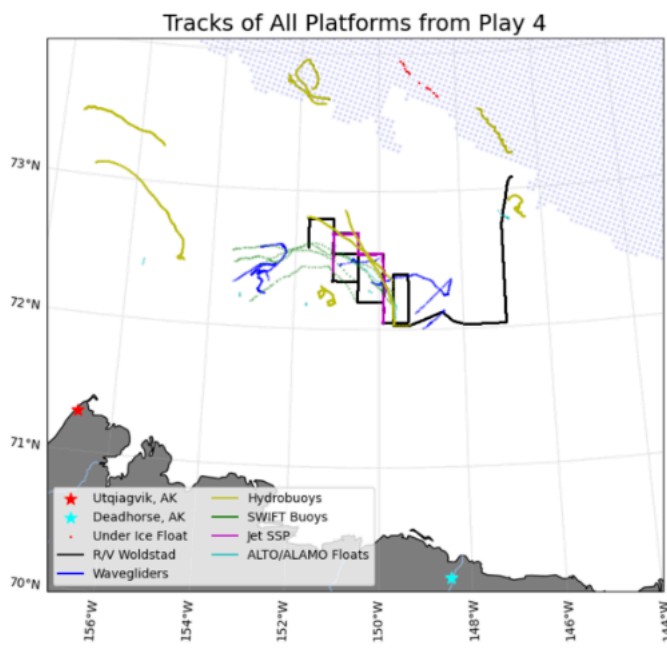


**Figure 48. A map showing the tracks of all six platforms during Play 4. Blue stippling indicates the presence of ice on 21 September 2022, as determined from the MASIE-NH ice product.**





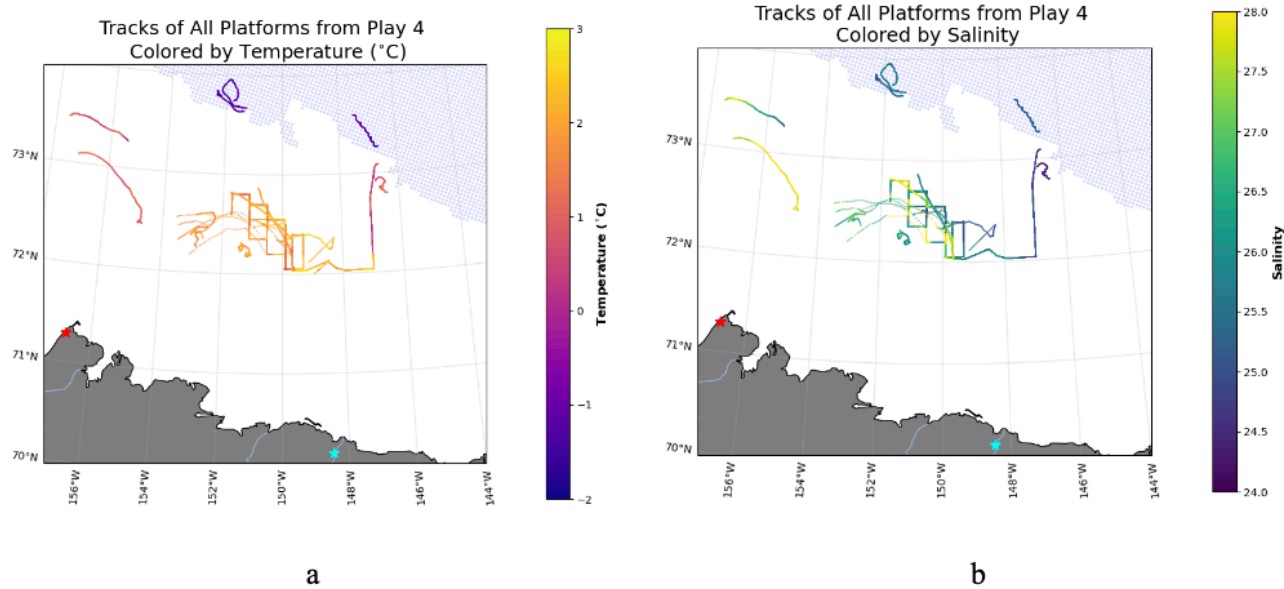

**Figure 49. Temperature (a) and Salinity (b) recorded closest to the sea surface by each asset during Play 4. The Under Ice float has been excluded. Blue stippling indicates the presence of ice on 21 September 2022, as determined from the MASIE-NH ice product.**

### 3.5 Play #5: Transects across ice-open water transition

The final play targeted the open water-sea ice transition immediately prior to freeze-up (Figure 50). Over four days, the ship made 14 consecutive transects along 150°30W collecting uCTD measurements that captured the evolution of the upper ocean as it moved toward the freezing point. Wave Gliders made north-south transects on either side of the ship, and measurements with drifting assets and JetSSP were also collected. Figure 51 shows a single section of uCTD data: notable features include a near-surface temperature maximum around 10-20 m depth that is maintained by the fresh surface layer; lateral variability above the pycnocline, with the presence of temperature fronts coinciding with the transition into sea ice; patchy Pacific summer water signals (warm water at 50-80 m depth) observed south of the ice edge; and vertical variability around 73.15°N.





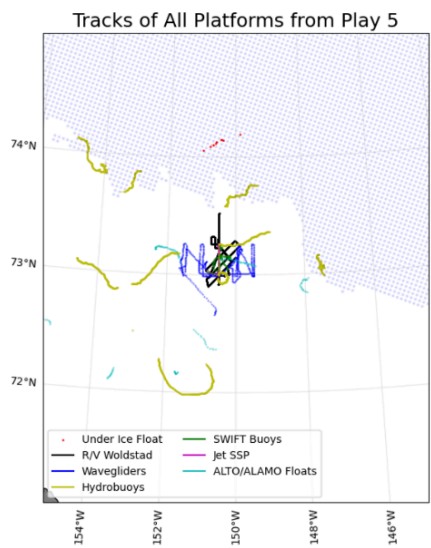

**Figure 50. A Map showing the tracks of all six platforms during Play 5. Blue stippling indicates the presence of ice on 27 September 2022, as determined from the MASIE-NH ice product.**



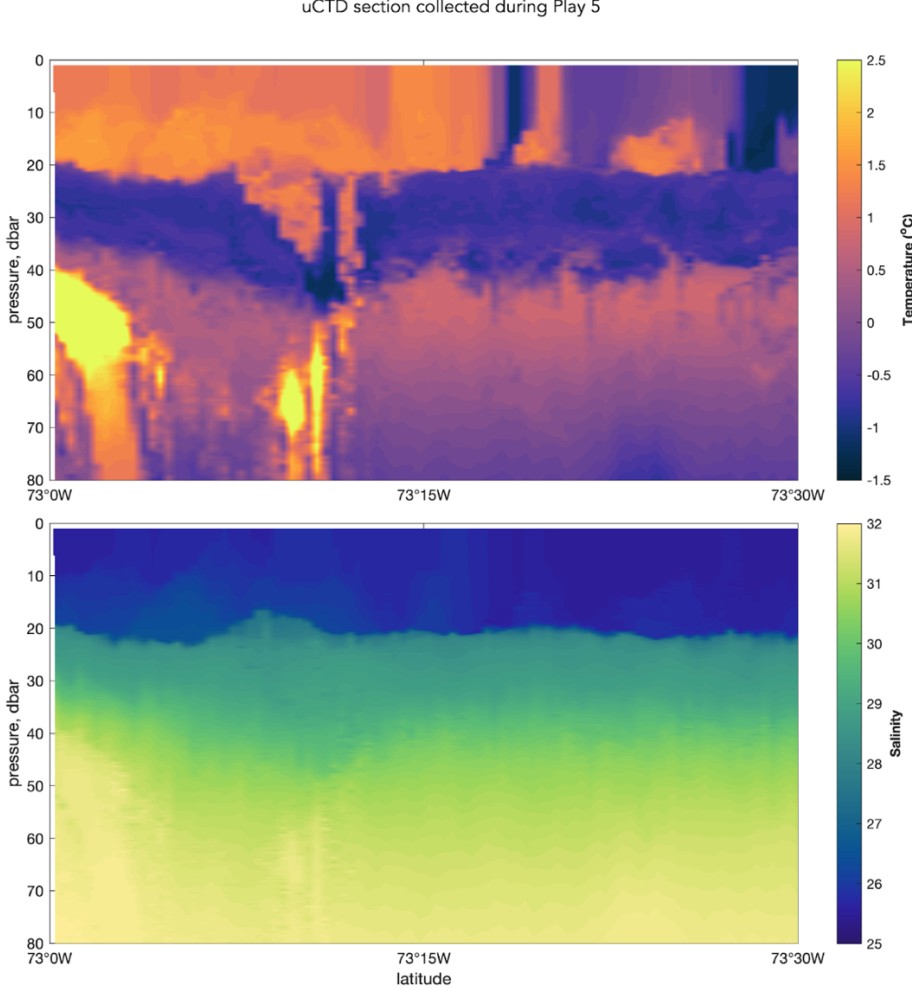

**Figure 51. Temperature and salinity collected with the uCTD along 150°30W during one section of Play 5 on 25 September. During this transect, sea ice was observed north of around 73.3°N (73°18'N), with open water observed to the south.**

## 4. Data Access

The following gives DOIs where SASSIE data can be accessed. All datasets are processed and calibrated, i.e. level 2 (L2), except where indicated. Data are accessed using https://doi.org/ [DOI], where [DOI] for each dataset is shown below.

| Dataset | DOI |
|---|---|
| Shipboard data | |



| TSG | 10.5067/SASSIE-TSG2 |
|---|---|
| Meteorology and air-sea flux | 10.5067/SASSIE-MET2 |
| Salinity Snake | 10.5067/SASSIE-SNK2 |
| ADCP | 10.5067/SASSIE-ADCP2 |
| Underway CTD | 10.5067/SASSIE-UCTD2 |
| Castaway CTD | 10.5067/SASSIE-RVCTD2 |
| S-band radar L3 | 10.5067/SASSIE-SBAND3 |
| $\delta^{18}O$ | 10.5067/SASSIE-D18O2 |
| Piloted/drifting data | |
| Wave Gliders | 10.5067/SASSIE-GLID2 |
| JetSSP | 10.5067/SASSIE-JETSSP2 |
| SWIFT | 10.5067/SASSIE-SWFT2 |
| HydroBuoys | 10.5067/SASSIE-HYDROGRAPHY2 |
| ALTO/ALAMO | 10.5067/SASSIE-PFLT2 |
| Under-ice float | 10.5067/SASSIE-ICFLT2 |
| Aircraft data | |
| PALS L1 | 10.5067/SASSIE-PALS1 |
| PALS L2 | 10.5067/SASSIE-PALS2 |

**Table 11. DOI for each of the published SASSIE datasets.**

## 5. Abbreviation Glossary

| Abbreviation | Stands for… |
|---|---|
| ADCP | Acoustic Doppler Current Profiler |
| AIS | Automatic Identification System |
| ALTO | A profiling float manufactured by Marine Robotic Vehicles |



| | |
|---|---|
| ALAMO | Air Launched Autonomous Micro-Observer |
| AMSR | Advanced Microwave Scanning Radiometer |
| APL | Applied Physics Laboratory of UW |
| CDOM | Colored Dissolved Organic Matter |
| CT | Conductivity and Temperature |
| C-FLUOR probe | Turner Designs fluorescence probe |
| COARE | Coupled Ocean Atmosphere Response Experiment |
| cCTD | Castaway Conductivity Temperature Depth Sensor |
| DCFS | Direct Covariance Flux System |
| DOI | Digital Object Identifier |
| ETOPO2 | National Centers for Environmental Information Topography and Bathymetry Global Relief Model |
| FDOM | Fluorescent Dissolved Organic Matter |
| GeoTIFF | Geographical Flag Image File Format |
| GPCTD | Glider Payload Conductivity Temperature, Depth Sensor |
| GPS | Global Positioning System |
| IMU | Inertial motion unit |
| JetSSP | Jet-driven Surface Salinity Profiler |
| MASIE | Multisensor Analyzed Sea Ice Extent (US National Ice Center, 2010) |
| MET | Meteorology |
| NASA | National Aeronautics and Space Administration |
| PALS | Passive Active L-band System |
| PO.DAAC | Physical Oceanography Distributed Active Archive Center |
| RBR | Name of a brand making submersible sensors for both physical and biological ocean variables. |
| RV | Research Vessel |



| RH | Relative humidity |
|---|---|
| SASSIE | Salinity and Stratification at the Sea Ice Edge |
| SBE | Sea-Bird Electronics |
| SeaCAT | Sea Bird CTD model |
| SMAP | Soil Moisture Active Passive |
| SSP | Surface salinity profiler |
| SSS | Sea Surface Salinity |
| SST | Sea Surface Temperature |
| SV3 | Surface Vehicle Model 3 |
| SWIFT | Surface Wave Instrument Floats with Tracking |
| TKE | Total Kinetic Energy |
| TSG | Thermosalinograph |
| uCTD | Underway Conductivity Temperature Depth Sensor |
| UIF | Under Ice Float |
| USV | Unmanned Surface Vessel |
| UW | University of Washington |
| VMDAS | Vessel Mounted Data Acquisition System |
| WHOI | Woods Hole Oceanographic Institute |

**Table 12. Glossary of abbreviations and acronyms used throughout the paper.**

Author contribution

KD and PG led the overall project. EW prepared the manuscript, figures, and code, with contributions from all co-authors. FMB and EW led the data publication. KD led collection and processing of the TSG, JetSSP, S-band Radar, and δ18O data.

PG and CS led collection and processing of the uCTD data. PG, EJR, and JT led collection and processing of the ADCP data. PG and JP led collection and processing of the cCTD data. SD and MS led collection and processing of the HydroBuoy data. SF and SM led collection and processing of the PALS data. VM and SZ led collection and processing of the meteorology and air-sea flux data. JT and EJR led collection and processing of the Wave Glider and SWIFT data. JJS led collection of the salinity snake data. AS led collection and processing of the under-ice float data.






**Code availability**

Code used to read the SASSIE data and generate the figures from this manuscript is available in Jupyter notebooks at https://github.com/lizwestbro/SASSIE_Data.

**Acknowledgements**

The SASSIE Program was supported by NASA Award 80NSSC21K0832. We are extremely grateful to the Captains and Crew of the RV *Woldstad* and to Support Vessels of Alaska, as well as the pilots of Kenn Borek Air, for enabling us to collect these data. We thank Léa Olivier and Laëtitia Parc for helping collect data on the SASSIE expedition.

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
