# Peer review of "Salinity and Stratification at the Sea Ice Edge (SASSIE): An oceanographic field campaign in the Beaufort Sea"

_Earth System Science Data, 2023_

## Author Comment (AC1)

**RC1 Response**

The paper presents details about SASSIE including instruments onboard the ship, the many uncrewed marine system assets, and the aircraft. General objectives of the study are discussed along with the sampling strategy employed for all assets deployed. Finally, the five major "plays" of the campaign are described where each one targeted a particular set of ocean conditions and regions. Data presented from each play indicate the vast amount of data that is available from SASSIE and how it will likely be used to understand links between salinity, near-surface stratification, upper ocean heat content, and sea ice freeze and melt. The paper presents a useful "catalog" of SASSIE data sets which will aid both the authors and future readers in using and interpreting the data. The few comments I have are listed below.

Thank you for taking the time to review our work. Please find responses to each of your comments below.

Figure 1: What does the color indicate (white vs. blue)?

The blue coloring indicates the extent of the ice field from the MASIE-NH National Ice Center Product. We have added this detail to the caption. (Line 78)

Line 118: Delete second "within the ice"

Deleted. Thank you. (Line 127)

Line 145: Just to clarify – the "previous ship and buoy computational fluid dynamics studies" refers to previous comparisons between ship and buoy measurements for a variety of ship-relative wind directions? A little more detail could be helpful here.

More details have been added to the 'wind speeds' part of this section (Lines 154-170):

"The ship's flow distortion was found to be a strong function of apparent wind angle, with a large blockage caused from the forward radar mast (roughly +150 degrees from bow for the port-side sensor, -150 degrees from bow for the starboard-side sensor). A linear average of the two wind speeds would effectively distribute the bias evenly between the two sensors. A study using multiple wind sensors on a buoy showed a similar bias pattern with apparent wind angle (Schlundt et al., 2020) and highlighted that the difference between the two sensors' wind speeds was distributed such that the upwind sensor accounted for roughly 75% of the difference, and the downwind sensor accounted for roughly 25% of the wind speed difference. We followed these percentages of (Schlundt et al., 2020), reducing the upstream sensor by 75% of the wind speed difference, and increasing the downwind sensor by 25% of the wind speed difference, with the error determined from the average percentage difference between the two wind sensors at relative direction from +100 to -100 degrees. For directions where one sensor was obstructed by the radar mast (near +/-150 degrees), this correction was not considered (since the nature of the flow distortion was different), and the unobstructed sensor data was used uncorrected."

The following reference was added to the paper:

Schlundt, Michael, et al. "Accuracy of wind observations from open-ocean buoys: Correction for flow distortion." *Journal of Atmospheric and Oceanic Technology* 37.4 (2020): 687-703.

Lines 150 – 154: There is not much quantitative information here, so the reader does not get a good sense of the accuracy of the blended temperature product. Could a time series comparison be provided so it is possible to see how the different instruments agreed or disagreed?

A time series of the temperature data is below. As noted, each of the three sensors had its own issues:  the Vaisala temperature was unstable (had numerous positive spikes) and therefore was excluded from the blended product, though its agreement with the Licor helped visually confirm the validity of the Licor. The Metek sonic temperature showed a mean bias compared to the other two sensors. The Licor showed a warm bias when the downwelling shortwave solar radiation was large (a known issue with unshielded temperature measurements).

[Figure]

At night, the Licor temperature and the Metek temperature agreed well (with a linear offset) indicating that they are usable data. We therefore used the Metek sonic temperature time series, and adjusted the linear offset to effectively "tune" the bulk sonic temperature flux to the directly measured sonic temperature flux.

The following was added to the text in the 'air temperature' section of the shipboard meteorology measurements (Lines 176-185).

'Sonic temperature measured from anemometers is known to differ from the true temperature due to the sensitivity of the speed of sound in air to water vapor. In most cases, sonic temperature is subject to sensor drift to a variety of factors, including the temperature of the sensor itself. For this deployment, the Metek sonic temperature was

relatively stable due to the generally low absolute humidity in cold temperatures, and because the sensor head was temperature controlled through a heating element. This lack of temperature drift was apparent through a high correlation between the night-time Licor air temperature (when the Licor sensor was not subject to radiative heating) and the Metek sonic temperature. Day-time air temperature values between the Licor and Metek were still correlated, however showed a bias with the Licor temperatures larger than the Metek temperatures consistent with radiative heating of the Licor sensor.. A corrected temperature product was created by finding the constant offset between the Metek Sonic temperature and the night-time Licor air-temperatures values to replace the biased day-time Licor air-temperature values. This corrected temperature product resulted in much better agreement between bulk and direct buoyancy fluxes than if any of the three raw air temperatures were used uncorrected.'

Figure 8: The uncertainties in b) and d) seem to be depicted by faint lines rather than dashes as indicated in the caption.

The caption has been changed to reflect the image (Line 210)

Figure 17: An indication in the text describing what the isotope data analysis shown in the figure reveals about the sources of freshwater during SASSIE would be helpful.

We have added a brief discussion about the isotope analysis results (Lines 370-374):

"$\delta^{18}$O values of -5 to -2 mL L$^{-1}$ were observed throughout the SASSIE domain, except for higher values (-2 to -1.5 mL L$^{-1}$) in the sea ice samples (Figure 17). While these values are generally consistent with previously observed $\delta^{18}$O values observed in Beaufort Sea upper halocline water and sea ice meltwater, and significantly higher than $\delta^{18}$O values of meteoric water (e.g., Lansard et al., 2012), further analysis is needed to quantify the sources of freshwater in the area."

---

## Author Comment (AC2)

**RC2 Response**

Review of "Salinity and Stratification at the Sea Ice Edge (SASSIE): An oceanographic field campaign in the Beaufort Sea" by K. Drushka et al submitted to Earth System Science Data

This manuscript presents a comprehensive overview of a field experiment ("SASSIE") that was carried out with NASA funding in the Beaufort Sea during the transition period from open water to freeze-up in the Beaufort Sea in 2022. A general overview is given of what the experiment was supposed to achieve. Then details on the different types of measurements are given. Finally, a run-down of the so-called "Plays" is given which are separate parts of the cruise that had different sampling strategies for distinct goals.

There are a number of detailed aspects (see below) which I think the authors could improve, but overall, the quality of presentation is good. However, there are two aspects about this manuscript which I find highly unusual. I explain them next but leave it up to the editor to judge how these fits into the scope and policies of the journal Earth System Science Data. Pending that, I would recommend major revision.

Thank you for taking the time to thoroughly review our work. Please find responses to each of your comments below.

1. I have seen many similar documents in my career before, but those were called "cruise reports". Those were not peer-reviewed but are still citable gray literature that can be identified through a persistent DOI. The major difference between what is submitted here, and a traditional cruise report is that also the DOIs of the final processed data sets collected during the cruise are given. As such this manuscript gives a good overview of the SASSIE data set and how the different sampling schemes in time, space, and between instruments compare. However, the description is minimal regarding the processing steps employed to get from the raw data acquired at sea to the final processed data.

My understanding up to now had been that the goal of ESSD is to provide a very detailed description of these processing steps. So, in that sense every individual data set provided with a DOI in this manuscript would be worthy of a separate ESSD paper. The submitted manuscript does not answer all (or even most) questions an end user of the processed data sets might have before they could use the processed data sets.

We understand the concern but feel that it is worthwhile to publish an overview of the SASSIE datasets in order to put each in the context of the broader SASSIE project and its associated data. This type of paper has been published in ESSD before: for instance, see Quinn et al. (2021) "Measurements from the RV Ronald H. Brown and related platforms as part of the Atlantic Tradewind Ocean-Atmosphere Mesoscale Interaction Campaign (ATOMIC)" (https://essd.copernicus.org/articles/13/1759/2021/), which provides a brief overview of the ATOMIC field campaign without extensive detail on the processing of each dataset.

2. I copy the start of the author contribution statement here:

"KD and PG led the overall project. EW prepared the manuscript, figures, and code, with contributions from all co-authors. FMB and EW led the data publication."

The author list reads "Kyla Drushka, Elizabeth Westbrook, Frederick M. Bingham, Peter Gaube et al" with Peter Gaube acting as corresponding author.

*Traditionally, the person who writes most of the manuscript (i.e. Elizabeth Westbrook in this case) should be first author and not the person who led the project.*

We recognize that this text was confusing. All co-authors contributed to the manuscript text. While EW made the figures and code and organized the text, KD led the project and wrote large portions of the text, including the introduction, campaign overview, five of the sections dedicated to specific platforms, and the summary of each of the five plays.

*Furthermore, based on the information provided in the manuscript, it does not become clear why Peter Gaube is the corresponding author. This should maybe made clear to the reader.*

PG took over the leadership role in SASSIE after KD moved to a different job. PG and will continue to fill this leadership role for the remainder of the project.

We have clarified the author contribution statement as follows (Lines 765-767): "KD and PG led the SASSIE project, with KD leading through the cruise and PG leading the post-cruise work. KD wrote sections 1, 3, and 4 of the manuscript. EW co-wrote the manuscript and prepared the figures, and code, with contributions from all co-authors."
* * *
**General comments:**

*The nature of the cruise report also shines through in the fact that the various tables provided employ a multitude of different formats, particularly with respect to how latitude/longitude are written. Dates/times should also be in a consistent format and in the sciences are provided in UTC; they may in addition (but not solely) be provided in local time which might be easier to interpret with respect to solar irradiation.*

All datetimes given in deployment and recovery tables have been put in the standard (ISO 8601) format 'yyyy-mm-ddTHH:MM:SSZ'. In tables that give information about the timeline of events (Tables 1, 9, and 10), the format used is 'dd Month' (ex: 10 September). All geographical coordinates given in the tables and text have changed so there is no space between the degree symbol and cardinal direction (ex: 72.2488°N)

*It is also not clear how complete the meta data provision is as it is not mentioned in the manuscript whether raw data, serial number, calibration sheets, and the like are archived at the same DOIs. Might be good to mention this in the manuscript.*

The following was added to the text in the 'Data access' section of the manuscript (Lines 751-755).

"All datasets have been quality controlled by the investigators who collected the dataset based on procedures appropriate for that dataset… Access to level 0 (L0) data can be obtained by contacting PO.DAAC at podaac@nasa.jpl.gov. Additional information about the Wave Glider, ALTO/ALAMO, and SWIFT datasets can be found at the DOIs below under 'Documentation'. All data files include CF-compliant metadata describing the instruments."

**Detailed comments:**

*L57 It might be helpful for the reader if the authors also provided links to how the relevant satellite data can be accessed in this manuscript.*

There are a range of satellite products that may be useful to the SASSIE project. Which datasets are considered relevant will depend on the particular question being asked, so we do not include links to specific datasets here; instead, we mention some of the typical satellite measurements that may be useful for the SASSIE project. This sentence now reads (Lines 61-63):

"Satellite measurements (for instance, sea surface salinity, temperature, and height; surface winds; sea ice concentration) are also a crucial part of the SASSIE mission data but are not addressed here."

L70 Figure 1 caption: "Sea ice extent" is not defined sufficiently. Do you show the 10% contour line? Comment on the fact that the satellite product may also claim that there is 0% somewhere when the concentration in fact is low ~5%.

The following definition was added to the caption (Lines 79-80):

"Sea ice extent" refers to the area where the MASIE-NH product indicates greater than 0 percent ice concentration."

ETOPO2: In the Arctic Ocean, typically IBCAO is used rather than ETOPO2.

Since the bathymetry is shown for reference only (and its accuracy is not central to interpreting the data), we feel that ETOPO2 is adequate and have opted not to change the plots to show IBCAO.

L80 See comments on local time vs. UTC above. Please provide UTC throughout

All tables have been updated so that the only place where local time is used is the timeline of events in table 1. This table does not give specific time stamps, only the days on which events occurred. This is made clear by adding "All other datetimes are in UTC." To the caption of table 1.

L103 Why were TSG data not logged at higher temporal resolution than once per minute?

Thanks for pointing this out. This was a typo; TSG data were logged every 10 s. The text has been updated. (Line 112)

Figure 8b/c why is positive into and out of the ocean in the different subplots rather than being consistent?

The figure was made like this to be consistent with the meanings of the data in the L2 meteorology files. We see the need for consistency here and have edited the figure so that positive always indicates fluxes into the ocean. (See new figure below)

[Figure]

 The following reference was added (Lines 848-849) :

'Ho, D. T., & Schanze, J. J.: Precipitation-Induced Reduction in Surface Ocean pCO₂: Observations From the Eastern Tropical Pacific Ocean. Geophys. Res. Lett, 47, https://doi.org/10.1029/2020GL088252, 2020.'

L208 This section lacks detail in comparison to the other sections.

We added a few more details to this section (Lines 244-255). However, because relatively little processing ultimately incorporated into the vertical shear dataset this section is relatively detailed.

L210 "the ship's" Please explain in more detail why it could not be removed correctly. What did you try? Give more details.

The ship's heading sensor did not function correctly, so accurate current vectors could not be recovered (Lines 246-248).

L212 Why do you not plot the vertical velocities as you plot any other data in this manuscript that ended up in the archived data collection?

The following figure showing the vertical shear collected at a sample depth of 24 m was added to this section (Figure 11).

[Figure]

Figure 12 Close-up photos of the instruments with a length scale indicated in the photos might be more effective in providing information to the reader about the collected data than the smiling people in the photos.

We have replaced the photos with close-ups and have added the length of the sensors to the caption (Figure 12).

L229 "the average depth of the casts was 100m with some reaching 200m".

"the average depth of the casts was 100 m" was changed to "the average depth of the casts was 100 m, with some casts reaching 200 m." (Lines 278-279)

L232 "quality controlled" There is no detail on what that entailed. Did you have salinity spiking issues? Did you correct them with a constant or varying drop speed? For the Teledyne Underway CTD, there are non-trivial processing steps as described at https://doi.org/10.1175/JTECH-D-13-00200.1 Was it necessary to consider these aspects?

The underway CTD data was processed via the RBR Ruskin software package. Data above 1 m were discarded, and only downcast data were retained. This information was added (Line 282)

The temperature sensor on the RBR concerto CTD is located forward of the conductivity cell, which helps to reduce salinity spiking in the data. However, we did observe salinity spiking near subsurface temperature maxima, and ongoing quality control work will focus on addressing this and removing it from the data.

L242 From the text it does not become clear whether the cCTD is a single use, expendable device or was used for multiple profiles.

The cCTD was used for multiple profiles. We feel that by adding a close-up photo of this instrument and referring to it as "the cCTD" instead of "a cCTD", it is clear that this is not an expendable sensor.

L245 "roughly 0.1" That is a very rough estimate…

Indeed: this is why we only used the cCTD as a last resort, when the risk of losing the much more accurate uCTD was too high.

L246 "quality control assessment" And what did you conclude from this assessment?

We revised this line (Lines 295-298) as follows: "Surface data from the cCTD were compared to TSG measurements to assess the accuracy of the cCTD measurements at the surface; profiles for which the cCTD temperatures were more than ± 2°C or the cCTD salinities were more than ±1.5 different from the nearest TSG measurements were discarded."

Figure 16 does not make the task of extracting information about the ice floes from the screenshots look promising.

We have included an image (Figure 16b) that shows a classified S-band radar data product, in which the image pixels have been classified to separate ice and no-ice. We also added a brief explanation of this product (Lines 341-348):

"A data product in which each pixel was classified as sea ice, not sea ice, sea clutter, or not data (e.g., range rings) was also developed (Drushka et al., 2023c). Pixels were classified based on their red, green, and blue (RGB) values: RGB values consistent with yellow to red color ranges were classified as "sea ice" and those consistent with blue-green colors classified as "no data". RGB values were summed as a function of distance from the center of each image: sea clutter appears as a peak in this value. An iterative method was used to identify the distance from center at which sea clutter could no longer be detected; pixels at smaller distances from center were flagged as clutter. The classified image corresponding to Figure 16a is shown in Figure 16b: though the classification is imperfect, it is evident that these data reveal qualitative information about the presence of sea ice around the ship."

L294 Should one not also have (accurate) salinity information to interpret the delta18O measurements? Please comment on how one will attain that.

Indeed, salinity data were collected with each bottle sample. This point is added to the text (Lines 364-368):

"Salinity measurements are needed to interpret $\delta^{18}O$ data, and salinity estimates collocated with each water sample are included in the $\delta^{18}O$ dataset. Each sample made with the lowered GoFlo bottle was collected within 5 minutes of a uCTD cast at the same location. Water samples made with the Salinity Snake were collocated with Salinity Snake data. Salinity of the sea ice samples estimated as a function of sea ice thickness (measured when the ice samples were made) based on Cox and Weeks (1974)."

L306 There is a Thomson and Girton (2017) paper and a Thomson et al (2018) paper in the references, but not a Thomson et al (2017) paper. It appears that you have not used a reference manager (not surprising given the varied list of authors who provided snippets of the manuscript), but then at least please carefully check your reference list against what is in the text.

Thomson et al. (2017) has been replaced with Thomson and Girton (2017) (Line 928). We have also gone through the reference list carefully.

L314 "within 10nm of the sea ice" How is that distance defined? 10% contour of satellite microwave product, which has low horizontal resolution and misses low ice concentration areas?

The following text was added (Lines 390-392): "(as determined from the NWS Alaska Sea Ice Program accessed at https://www.weather.gov/afc/ice, an operational product that ingests visible imagery to help include low ice concentration areas)."

L322 Should the RMS be different, or do you expect a true vertical difference (related to the mean salinity stratification) of that magnitude (0.1)? Why "(not shown)"?

The RMS difference of 0.1 is consistent with the accuracy of the inductive conductivity cells used on the Wave Gliders. It is expected. The scatter plot is not shown here purely for brevity.

Table 4 Aanderaa CT sensor was at 0.2m according to Table 4 and at 0.25m according to the caption of Figure 20!? Also the other depths (1m, 4m, 8m): Please provide them with the same number of significant digits, i.e. 1.0m, 4.0m, 8.0m or 1.00m, 4.00m, 8.00m depending on whether 0.2m or 0.25m is correct. Also compare L348.

We have updated the table as follows: 0.25 changed to 0.25 m, 1.0 changed to 1.00 m, 4m changed to 5m, 8 m changed to 10 m. Because the CT sensors at 5 m and 10 m were on flexible tethers so their depth is not stated with as many significant digits.

Figure 23 What depth are the plotted data from?

 The caption was changed to (Lines 435-437): "Figure 23. Tracks of JetSSP deployment 4 on 09-20-2022 color-coded by (a) temperature and (b) salinity taken at 0.05 m depth. The thin lines show the track of the RV *Woldstad*. Black circle marks initial deployment location and black star marks the recovery."

L359 Was it intended to be lost? Otherwise, it is a self-evident statement that it was the final one.

 This sentence was changed to 'On the eighth deployment, the platform was lost.' (Line 442). This was meant to be the final deployment.

L367 "SWIFTs" Please provide the full name in addition to this abbreviation.

The first sentence of this section (Line 449) has been changed to "Five Surface Wave Instrument Floats with Tracking (SWIFTs; Thomson 2012) were repeatedly deployed and recovered in the Beaufort Sea during the SASSIE field campaign (Figure 26)."

Figure 27a Please describe what the yellow thing is that is located in the left of the photo.

The following was added to the caption (Lines 459-460): "Visible in (a) is a small yellow float and line that trails behind the drifting SWIFT to make it easier to recover with a grappling hook."

L383 "offset of almost 1" So have you removed that data from the final data set? Please clarify in the text.

Yes, that data was removed as part of QC before publishing the data. We have added the text, "this offset was removed from the final dataset". (Line 468)

Figure 28 would be more useful to the reader if you commented on the north-south extent in km of the line along which the deployments took place. Also you would need to specify of how many days of data (at least typically) the shown tracks are comprised.

We added the following text: "For instance, during play 1, five SWIFTs were deployed along a 18 km meridional line and allowed to drift for ~ 2 days. During play 4, four SWIFTs were deployed across a 40 km long line." (Lines 478-480) The deployment and recovery dates for each of the SWIFT deployments are now shown in Table 6 to give the reader information about how long the deployments were.

L408 "several equipped with a drogue" Table 7 hopefully indicates the presence/absence of a drogue and the depth of the drogue.

Table 7 has been updated to include whether buoys were drogued, and the text was updated to read "six were equipped with a drogue consisting of "holey sock" fabric cylinders that are 1 m in diameter and 6 m in length, centered at 15 m depth." (Lines 498-500)

Table 7 Deployment time first two buoys: It is hard to believe that you deployed 2 buoys exactly at midnight UTC (00 seconds). Or is it correct that they had an initial horizontal separation in the water of 0m?

Table 7 has been updated to reflect the deployment time of each buoy.

Table 7 Have all of the buoys stopped recording data as of the time of the submission of the manuscript or are some still recording? Consider mentioning in the table.

The following was added to the caption of Table 7: "As of March 2024, HydroBuoys 002, 007, 008, and 009 were still recording." (Lines 506-507)

L438 What type of Seabird CTD was used?

The float is equipped with two SBE 41CP units (this information is now added to the text). (Line 529)

L445 "frequent" How frequent?

The frequency of the profiles varied through the course of the mission, but it was typically between 12 and 18 profiles per day (15 profiles/day on average). Added to the text. (Lines 546-547)

L453 Did the float have a parking depth or did it continuously profile?

The float profiled continuously and did not have a parking depth. It did, however, pause periodically (every 3rd profile) at 15-20m depth for about an hour to survey the underside of the ice and take photos. The float also paused at the surface approximately 4 times a day to transmit the data (whenever open water was found).

L480 "drift laterally at depth of XXm".

The floats drift laterally at all depths throughout their profile (upper 100 m). This sentence has been changed in the text (Line 576)

L488 Please explain the ice avoidance algorithm in more detail to refer to where it is described in more detail.

A reference for the ice avoidance algorithm was added (Wong et al. (2020)).

Table 8 Why do you only provide one decimal digit for some of the longitudes/latitudes? You should know the position better than to within 6nm!

Latitude and longitude of deployments and final transmissions are now given to the fourth decimal place in table 8.

L572 How heavy?

Added "estimated to be >60% ice cover" to the text (Line 675)

L578 "the presence" What % concentration does that refer to?

The captions for all figures in Section 3 has been changed to "Blue stippling indicates an ice concentration greater than 0%"

L594 "has been excluded" Why?

 "The under-ice float has been excluded because it does not consistently make measurements near the surface" was added to the caption of Figure 44. (Lines 684-685)

L613 "signals of interest" What were those?

"signals of interest" has been replaced with "strong salinity gradients".  (Line 721)

L614 That sounds really interesting, however Figure 49 does not provide the reader with much they could interpret with respect to this point.

 Images 'c' and 'd' were added to figure 50 to show the up-close salinity and temperature gradients in the drifter following boxes.

"A zoomed in view of the temperature (c) and salinity (d) recorded by the drifter following boxes is also shown."

Was added to the figure 50 caption. (Line 729)

L629 "and vertical variability around 73.15°N" What do you mean by that?

We agree this statement was unclear and have deleted it from the text.

Figure 51 Latitude is "N", not "W" on the x-axis. It might be nice to have enough x-ticks such that also 73.3°N (the ice edge) is included and the reader does not have to visually interpolate along the x-ticks to find the impact of the sea ice edge in the temperature data.

These changes have been made.

L638 "All datasets are processed and calibrated." I would have expected more detail on those steps in this manuscript.

We added citations to Table 11 pointing to processing/quality control for each dataset, if available.

Table 12 The descriptions of what "RBR" and "SBE" are inconsistent. WHOI: Institute -> Institution

We deleted RBR from Table 12 because it is not an acronym. We have replaced "Institute" with "Institution".

L653 The other co-authors led the "collection and processing", but JJS only led the "collection"?

This has been updated to "JJS led collection and processing of the salinity snake data." (Lines 773-774)
* * *
**These minor typing and grammar errors have been addressed. Thank you for the very careful reading of the manuscript.**

L28 inducing -> introducing

L39 arctic -> Arctic

 L75 "Woldstad" is italic in the caption of Figure 2, but not in Figure 1. Please be consistent throughout the manuscript.

L282 "north is up" info is repeated just half a sentence later.

L288/289/573 Make sure that the "18" is in proper superscript. Currently it's different in the two lines, but not correct in either.

Table 2 Sensor depth of Licor is missing unit.

Table 3 Sometimes there is a space before the degree sign and sometimes there is not. "71.08 °N" and "71.12°N". The format for latitude and longitude is also different than e.g. the caption of Figure 15, which is also different than the format in the title of Figure 15.

L319 KHz -> kHz

Table 7 What happened to the horizontal line on the bottom of the second column of the table on both pages?

L423 "86°N"

L400 "during the SASSIE cruise" This snippet of text was clearly written by someone else, because elsewhere in the manuscript this is always written as "by the RV Woldstad".

L661 "extremely grateful". Maybe "grateful" would suffice.

L708/711 The links are only partially underlined. Throughout the reference list decide to either have the links underlined or not. Currently some are and some other are not.

Table 5 third column features "N, 1", "N ,1" and "N 1". These are basic (of course they are tedious…) editing steps that should be done before the submission of a manuscript.

L549 "GHz," -> "GHz"

L574 "gradients of 2-3" -> "gradients of 2-3 / XXkm" or "differences of 2-3"
* * *
**The following figure related changes have been made:**

The font sizes on almost all figures are too small. They need to be easily legible when printed out in letter/A4 size. Print it out and try to read it yourself. Given that the figures are all made with the same script, this should be an easy fix.

The font sizes have been increased on the colorbar and latitude/longitude labels on almost all figures.

Figure 7 is a weird projection in which it is very hard to make sense of the direction vectors. How about Mercator?

Figure 7 has been changed to a Mercator projection.

Figure 9a Add a line with the freezing temperature as calculated from the salinity.

A blue line showing the freezing point calculated from salinity measured by the salinity snake has been added to the figure. 'The blue line indicates the freezing temperature of the water.' has been added to the caption.

Figures 13/14 Please use the same x- and y-limits and color scales in both figures to make it possible for the reader to compare the two.

The color scales have been changed so that they match.

Figure 20 title "Readings" Is that a technical term?

The title of these figures has been changed.

Figure 24 How are the x-tick labels "2022-09-20" and "2022-09-20" meaningful?

The center label has been removed so that the labels now only read '2022-09-20' and '2022-09-21'

L469 (figure 35) "Note the different" I cannot read either one given the font size. Also, please keep them the same to give the reader a chance to compare the two.

The font size on these two images has been updated and the color bars have been set to the same scale.

Figure 39 An alternative way to display this would be a map of the trajectory and a depth vs. time contour plot of the temperature/salinity.

The 3D grid contour plot has been changed to a depth vs. time section, and the trajectory shown in Figure 38 has been annotated with a start to indicate the path belonging to float 9098.

Figure 8d Could also add a 24 hour moving window.

We are not inclined to do any smoothing on this data in the manuscript because we want it to be an accurate reflection of the data that is archived in the L2 files.